# A dopaminergic switch for fear to safety transitions

Ray Luo[1,2], Akira Uematsu[1,2], Adam Weitemier[1], Luca Aquili[1,4], Jenny Koivumaa[1,2]
Thomas J. McHugh [1,2,3] & Joshua P. Johansen [1,2,3]

Overcoming aversive emotional memories requires neural systems that detect when fear responses are no longer appropriate so that they can be extinguished. The midbrain ventral tegmental area (VTA) dopamine system has been implicated in reward and more broadly in signaling when a better-than-expected outcome has occurred. This suggests that it may be important in guiding fear to safety transitions. We report that when an expected aversive outcome does not occur, activity in midbrain dopamine neurons is necessary to extinguish behavioral fear responses and engage molecular signaling events in extinction learning circuits. Furthermore, a specific dopamine projection to the nucleus accumbens medial shell is partially responsible for this effect. In contrast, a separate dopamine projection to the medial prefrontal cortex opposes extinction learning. This demonstrates a novel function for the canonical VTA-dopamine reward system and reveals opposing behavioral roles for different dopamine neuron projections in fear extinction learning.

[1] RIKEN Brain Science Institute, 2-1 Hirosawa, Wako-shi, Saitama, Japan 351-0198. [2] RIKEN Center for Brain Science, 2-1 Hirosawa, Wako-shi, Saitama, Japan 351-0198. [3] Department of Life Sciences, Graduate School of Arts and Sciences, University of Tokyo, Tokyo, Japan. [4] Present address: Department of Psychology, Sheffield Hallam University, Sociology & Politics, Heart of the Campus Building, Collegiate Crescent, Collegiate Campus, Sheffield S10 2BQ, UK. These authors contributed equally: Ray Luo, Akira Uematsu. Correspondence and requests for materials should be addressed to J.P.J. (email: jjohans@brain.riken.jp)

Exposure therapy, a form of extinction learning, is an important psychological treatment for anxiety disorders such as post-traumatic stress disorder (PTSD). Extinction of classically conditioned fear responses is a model of exposure therapy. In the laboratory, animals learn that a sensory stimulus predicts the occurrence of an aversive outcome through fear conditioning. During extinction, the omission of an expected aversive event signals a transition from fear responding to safety. To switch from fear responding to extinction learning, a brain system that recognizes when an expected aversive event does not occur is required. While molecular changes occurring in the ventromedial prefrontal cortex (vmPFC) and amygdala are known to be important for storing and consolidating extinction memories[1,2], the brain mechanisms for detecting when an expected aversive event did not occur and fear responses are no longer appropriate are less well understood[3].

One candidate circuit for this function is the midbrain ventral tegmental area (VTA)-dopamine system. VTA-dopamine neurons signal reward through phasic responses to better-than-expected outcomes. They are more strongly activated by unexpected, compared with expected, rewards and inhibited when an expected reward does not occur, thereby encoding reward prediction errors[4–6]. The omission of an expected aversive outcome, as occurs during extinction is also known to activate dopamine neurons and increase dopamine levels in the nucleus accumbens (NAc)[7–11], particularly in the NAc medial shell (mShell)[10]. Dopamine neurons are also activated by the cessation of ongoing aversive stimuli and activity in VTA-dopamine neurons and their projections to the NAc are important in relief learning, a process which may be similar to extinction[11–13]. Dopamine neurons are heterogeneous in their response to rewarding and aversive outcomes[7,8,14], with some cells responding specifically to reward and others more generally to salient stimuli. A recent study in *Drosophila* also reported that a distinct population of dopamine neurons are involved specifically in extinction of reward learning based on their connectivity with a specific area of the mushroom body[15]. In mammalian systems, separable populations of dopamine neurons project to the NAc, amygdala and mPFC[16] and distinct calcium changes have been reported in different VTA-dopamine terminal fields[17,18]. Importantly, dopamine receptor activation in the vmPFC, amygdala and NAc modulates extinction learning and memory[19–23]. This suggests that based on their projection targets, distinct subpopulations of dopamine neurons may differentially regulate fear extinction. What is unknown is whether activation of VTA-dopamine neurons during the omission of an expected aversive outcome is necessary for fear extinction learning and, if so, whether this effect is mediated by specific populations and projections of dopamine neurons.

Using a circuit specific optogenetic-behavioral approach, we show that optogenetic inhibition of VTA-dopamine neurons during expected shock omission is necessary for extinction learning. Furthermore, we report that specific populations of VTA-dopamine neurons project to the NAc mShell or core and inhibition of the mShell projecting cells reduces the long-term retention of extinction. In contrast, a separate projection of the VTA-dopamine system to the vmPFC opposes extinction learning. Together, these results demonstrate that the VTA-dopamine system is necessary for detecting when aversive responses should be reduced and reveal that distinct populations of dopamine neurons make unique and specific contributions to this process.

## Results

**VTA-dopamine activity switches fear responding to extinction learning**. We first examined whether activity in VTA-dopamine neurons during the shock omission period of fear extinction was necessary for extinction learning. To do this we expressed enhanced yellow fluorescent protein (eYFP) alone or the inhibitory Halorhodopsin fused to eYFP (eNpHR3.0-eYFP) specifically in VTA-dopamine cells of tyrosine hydroxylase-cre recombinase (TH-Cre) rats[24] by injecting a Cre-dependent adeno-associated virus (AAV) into the VTA (Fig. 1a, b and Supplementary Fig. 1). Optical silencing of VTA-dopamine neurons and dopamine release using NpHR has been demonstrated previously, providing validation of our approach[25,26]. Animals then underwent fear conditioning during which a neutral auditory conditioned stimulus (CS) was paired with an aversive footshock unconditioned stimulus (US). This was followed 24 h later by presentation of the auditory CS without the aversive US during an "extinction" learning session (Fig. 1a) in which animals first show behavioral freezing responses that gradually diminish over the course of extinction. 24 h later an extinction "retrieval" session occurred in which animals were presented with the CS alone again to determine their retention of extinction memories. During extinction trials, illumination of VTA occurred during either the shock omission period (Fig. 1a, c and Supplementary Fig. 2a) or during each auditory CS period (Fig. 1a, d and Supplementary Fig. 2b) of extinction training in animals expressing eNpHR3.0 or eYFP. While eYFP controls and animals in which optical inhibition occurred during the auditory CS period reduced their freezing responses during extinction, extinction learning and subsequent retrieval were significantly reduced in animals with optical inhibition occurring during the expected shock omission period (Fig. 1c, d). There were no detectable differences between eYFP and NpHr animals in freezing responses prior to the onset of the first CS (% freezing in 20 s before 1st CS onset: NpHR = 2.7 ± 1.8% SEM, eYFP = 5.6 ± 2.0% SEM; $p = 0.33$, student's t-test). To determine whether extinction was completely blocked we used a more specific analysis comparing freezing during the first CS of extinction to the last CS of extinction and to the first CS of extinction "retrieval". This showed that compared with freezing to the first CS of extinction training, there was a significant reduction in freezing by the final CS of the extinction training session and this reduction persisted to the first CS of extinction retrieval (Supplementary Fig. 3a). Together this shows that although laser application produced a large attenuation of extinction learning during training, some extinction did occur in eNpHR3.0 treated animals. Notably, pairing an auditory CS with optical inhibition of dopamine neurons alone, in the absence of shock, was not sufficient to produce fear conditioning (Supplementary Fig. 3b, c). Combined with the data showing normal extinction when inhibition occurred during the entire CS period (rather than during the shock omission period) of extinction, this suggests that inhibition of dopamine neurons did not reduce extinction by acting as an aversive stimulus in place of the shock. Together, these data demonstrate that dopamine activity during the expected shock omission period, but not during the auditory CS period is necessary for normal extinction learning to occur.

Phosphorylation of MAP kinase (pMAPK) occurs in the amygdala and vmPFC and is necessary in these regions for normal extinction learning[27–29]. We next investigated whether inactivation of VTA-dopamine neurons during the shock omission period reduced extinction learning induced increases in pMAPK[27–29]. We expressed eNpHR3.0 or eYFP in VTA-dopamine cells, optogenetically inhibited these cells during the shock omission period of fear extinction and used immunohistochemistry to quantify MAPK phosphorylation (pMAPK). We compared this to chamber exposed control animals who were fear conditioned but did not undergo extinction training (Fig. 1e). In the eYFP treated groups, extinction training increased pMAPK levels in lateral amygdala (LA) and infralimbic (IL) subregion of mPFC compared to chamber exposed controls (Fig. 1f–h). In

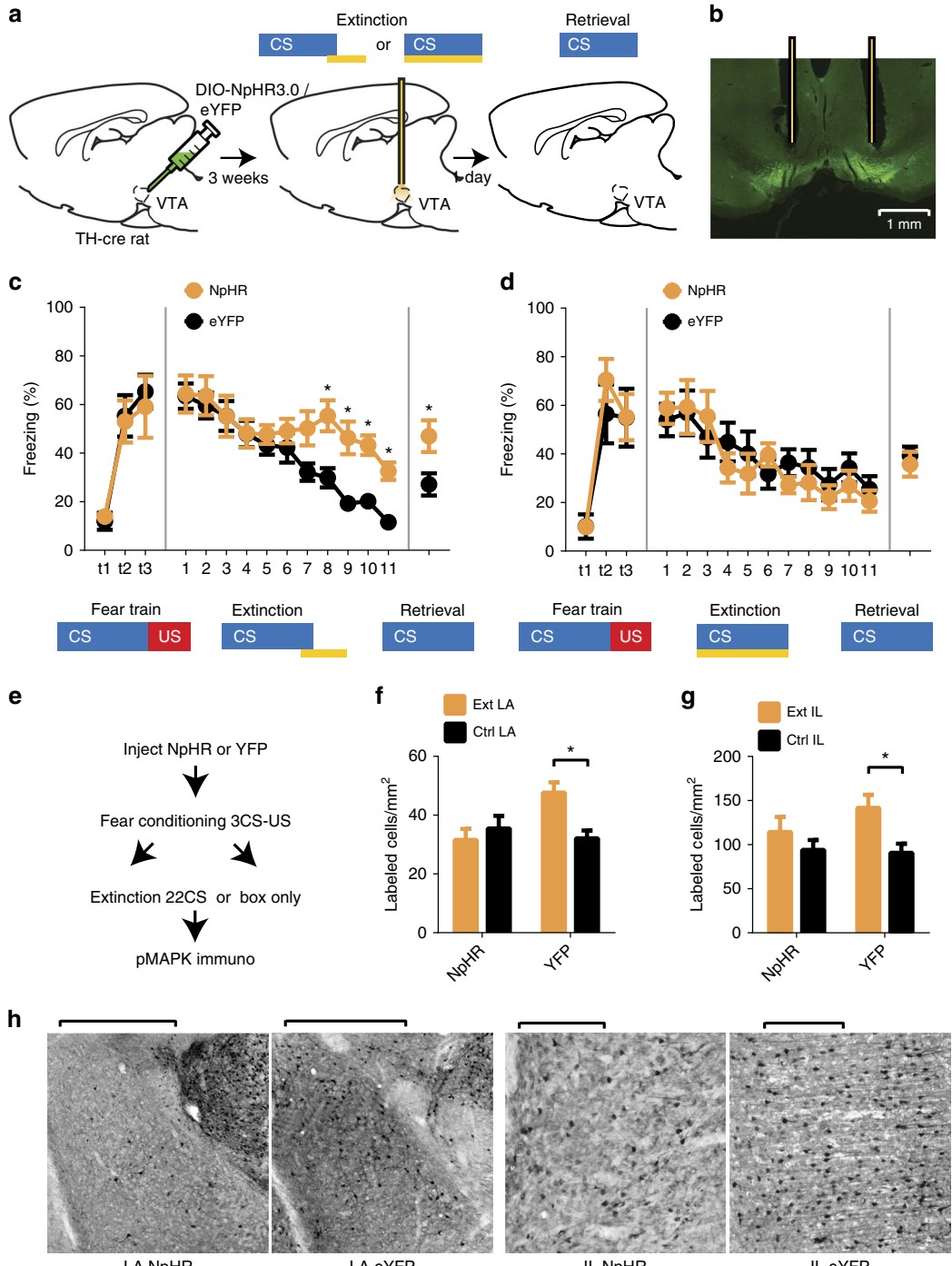

**Fig. 1** Optogenetic inhibition of VTA-dopamine neurons during shock omission blocks extinction learning. **a** Experimental paradigm for optogenetic inactivation of VTA-dopamine neurons during expected shock omission or auditory CS presentation of fear extinction. Brain image adapted from ref.[70]. Copyright 1982, Elsevier. **b** Expression of NpHR3.0-eYFP in VTA (scale bar = 1 mm). **c** Inactivation of VTA-dopamine cells during shock omission period reduced "extinction" learning ($n = 7$ NpHR, 10 YFP, 2-trial average, $F_{10, 150} = 4.709$, $p < 0.0001$ significant interaction, 2-way repeated measures ANOVA, *$p = 0.0090$, 0.0050, 0.0229, 0.0466 for trials 8, 9, 10, 11, simple effects of inactivation, Holm–Sidak multiple comparison) and extinction memory consolidation measured during a "retrieval" test (5-trial average, *$p = 0.0212$, Student's $t$-test). **d** Inactivation of VTA-dopamine cells during CS tone presentation affected neither extinction learning ($n = 6$ NpHR, 6 YFP, 2-trial average, $F_{10, 100} = 0.903$, $p = 0.5339$ no interaction, 2-way repeated measures ANOVA) nor consolidation (5-trial average, $p = 0.5874$, Student's $t$-test). **e** Paradigm for assessing MAPK phosphorylation (pMAPK) in lateral amygdala (LA) and infralimbic (IL) cortex following optogenetic inhibition of VTA-dopamine cells during shock omission period of extinction. **f, g** Inhibition of VTA-dopamine neurons during shock omission period of extinction abolished the increase in pMAPK following extinction found in non-inhibited animals (LA: $n = 8$ NpHR, 10 YFP, 4-slice average, $F_{1, 32} = 7.153$, $p = 0.0117$ significant interaction, $p = 0.0333$ main effect of inactivation in extinction, *$p = 0.0287$ simple effect of extinction in YFP; IL: $n = 8$ NpHR, 10 YFP, $F_{1, 32} = 1.343$, $p = 0.255$ no interaction, $F_{1, 32} = 7.334$, *$p = 0.011$ simple effect of extinction in YFP; 2-way ANOVA). **h** Representative examples (left LA, scale bars = 1 mm, right IL, scale bars = 0.2 mm) of pMAPK positive cells in extinction groups in NpHR3.0-eYFP and eYFP treated animals. Error bars indicate SEM

contrast, optogenetic inhibition of VTA-dopamine cells in the eNpHR3.0 treated group during shock omission prevented the increase in pMAPK associated with extinction learning in the amygdala and IL. This shows that activation of dopamine neurons during shock omissions in extinction training promotes molecular changes in LA and IL that support extinction learning.

**Inhibiting shock activity in dopamine neurons enhances fear.** Electrophysiological studies of the dopamine system have suggested that one function of dopamine could be to signal the occurrence of any salient event[7,30]. While activation of VTA-dopamine neurons is necessary during rewards for learning to occur and VTA-dopamine signaling has been implicated in aversive learning[31–33], the role of shock-evoked activity in dopamine neurons in fear learning is not clear. We tested this question using the same approach described above to inhibit the activity of VTA-dopamine neurons during the shock US period of fear conditioning (Fig. 2a). We found that optogenetic inhibition of VTA-dopamine neurons during the shock ("NpHR" group) enhanced fear learning compared with eYFP expressing animals or NpHR expressing animals that received laser inhibition after the shock had already occurred (Offset group, Fig. 2b). Furthermore, inhibition during the auditory CS period of conditioning had no effect on learning (Fig. 2c). Combined with the findings above that inhibition of dopamine neurons alone paired with shock does not produce fear learning (Supplementary Fig. 3c), this suggests that neural activity in dopamine neurons during the shock period serves to restrain fear learning. However, although global inhibition of VTA-dopamine neurons enhanced fear learning, specific dopamine projections may individually facilitate fear learning[34].

**Different dopamine projections enhance or reduce extinction.** VTA-dopamine neurons are heterogeneous based on their differential projections to distinct efferent targets[14,16–18]. To test whether different VTA-dopamine neuronal projections serve distinct functions for fear extinction learning, we expressed the inhibitory opsin ArchT-GFP in VTA-dopamine cells and examined the effect of inactivating the terminals of these cells in various dopaminergic target regions. We used ArchT here because NpHR3.0 did not express well in rat dopamine nerve terminals/axons and because previous studies have demonstrated the efficacy of inhibiting terminal release with archaerhodopsin using short time duration laser illumination ($< 60$ s)[35,36]. To verify that optogenetic manipulation of dopamine nerve terminals inhibited dopamine release, we used fast-scan cyclic voltammetry (FSCV) to measure dopamine release in NAc (Fig. 3a and Supplementary Fig. 4a–c). We chose the NAc to confirm optogenetic control of dopamine release because it is a well-established site for dopamine measurement[37–39]. Also, among the neuromodulators detectable by FSCV, MFB stimulation elicits predominantly dopamine release within the NAc[40]. We found that evoked dopamine release in the NAc was attenuated by light inhibition in ArchT animals, but not in GFP control animals or in ArchT animals without laser illumination (Fig. 3b, c and Supplementary Fig. 4d, e). The observed inhibition was limited, likely because dopamine fibers were stimulated electrically which would strongly activate most dopamine fibers, many of which may not be expressing ArchT. We then investigated the effect of optogenetically inhibiting dopamine terminals in different projection targets during the shock omission period of extinction training. Consistent with evidence of increased dopamine release in the NAc during omission of expected aversive outcomes[9,10], we found that inhibition of dopamine terminals in NAc during extinction training impaired retention of extinction memories

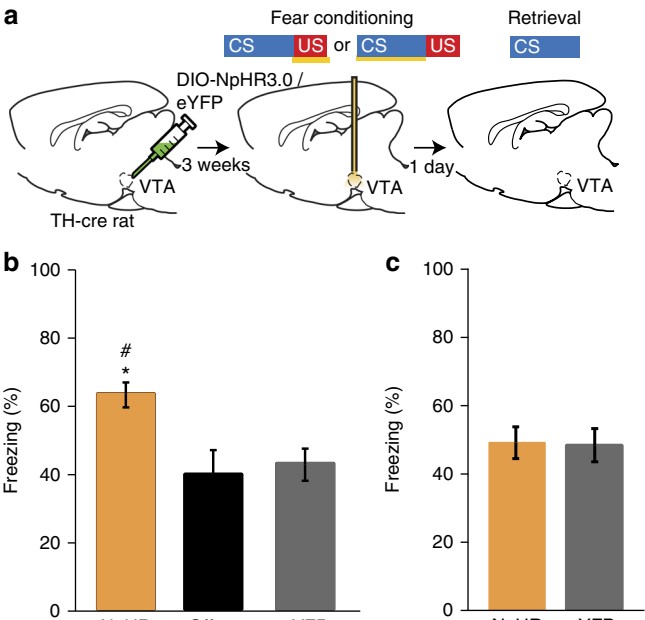

**Fig. 2** Optogenetic inhibition of VTA-dopamine neurons during shock period of fear conditioning enhances learning. **a** Experimental paradigm for optogenetic inactivation of VTA-dopamine neurons during shock or auditory CS period of fear conditioning. Brain image adapted from ref.[70]. Copyright 1982, Elsevier. **b** Inactivation of VTA-dopamine neurons during shock US period enhanced fear conditioning ("NpHR", $n = 16$), relative to "Offset" ($n = 14$) and 'eYFP' ($n = 10$) control groups ($F_{2, 37} = 6.4$, $p = 0.0041$, * and # denote significant differences between NpHR and Offset as well as NpHR and eYFP groups using Holm–Sidak posthoc tests) when comparing behavioral freezing levels (% freezing during auditory CS) during the retrieval test period 24 h after conditioning. **c** Inactivation of VTA-dopamine neurons during auditory CS period of fear conditioning ("NpHR", $n = 8$) did not change fear conditioning relative to "eYFP" control ($n = 10$) ($p = 0.92$, Student's $t$-test). Error bars indicate SEM

(Fig. 3d, e and Supplementary Fig. 5a). In contrast, inhibition of dopamine nerve terminals in the vmPFC enhanced extinction memory consolidation/retention (Fig. 3f, g and Supplementary Fig. 5b) while inhibition of terminals in the lateral/basal amygdala had no effect (Fig. 3h, i and Supplementary Fig. 5c). To determine the extent to which extinction was affected by NAc terminal inactivation we again compared the first CS of extinction with the last CS of extinction as well as with the first CS of extinction retrieval in the ArchT treated group. We found a significant reduction in freezing comparing 1st to last CS of extinction training, but freezing completely recovered 24 h later during the retrieval test (Supplementary Fig. 5d). We also found no differences in baseline freezing levels between ArchT and GFP expressing animals prior to the onset of the first CS of extinction retrieval test (Supplementary Fig. 5e). Together this shows that specific projections of dopamine neurons serve opposing roles in fear extinction learning, with NAc projections facilitating and vmPFC projections opposing long-term retention of extinction memories.

**NAc-mShell projecting dopamine cells regulates extinction.** Previous work has shown that release of dopamine in the NAc core and mShell can be decoupled during different behaviors and in response to distinct stimuli[10,41]. Notably, increases in dopamine release when an expected shock is omitted occurs in the mShell[10], but dopamine release is reduced in core in response to aversive predictive cues[9,10]. This suggests either that distinct

populations of dopamine neurons project to core and mShell or that terminal release of dopamine is uniquely modulated in each of these regions. Retrograde tracing studies have suggested that dopamine inputs to these NAc subregions may arise from different parts of the VTA[42], but no study has directly tested this question using a multi-tracer approach. To do so we injected two different colored retrograde tracers into the NAc core and mShell and quantified the degree of overlap in VTA neurons labeled with these differently colored tracers. We found completely non-

overlapping VTA dopaminergic cell populations projecting to NAc core and mShell (0% overlap, $n = 4$ animals, Fig. 4a) explained partially by a medial (dominated by mShell projecting cells) to lateral (dominated by core projecting cells) segregated, topographic organization as has been suggested by previous work[42], but also by distinct cell populations in VTA areas where intermingled cells were apparent.

Next we examined whether inhibition of mShell or core projecting VTA-dopamine neurons during the shock omission

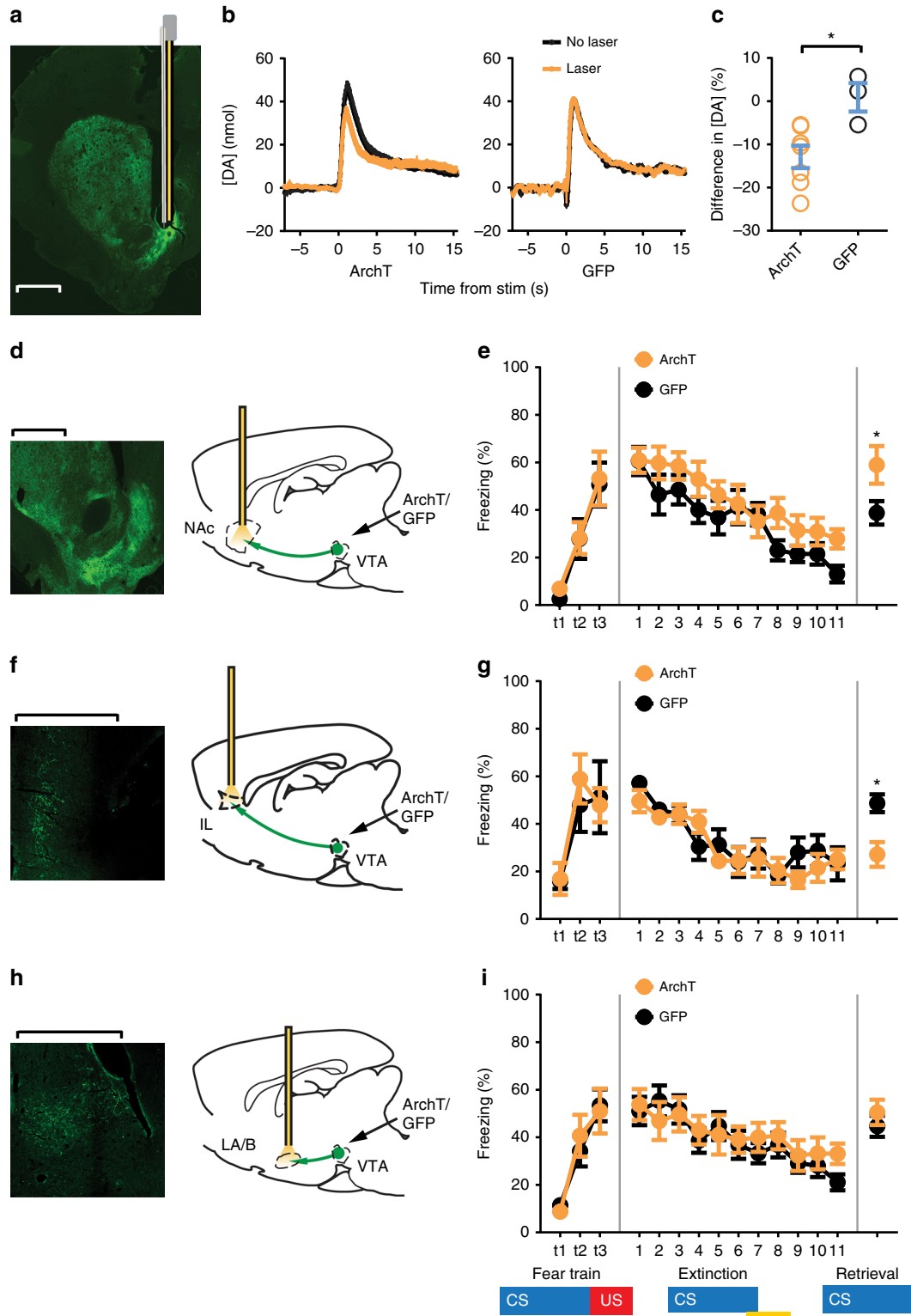

period of extinction affected behavioral extinction. Because some retrogradely labeled neurons were not TH-positive/dopaminergic we used a combinatorial retrograde infection approach that targets opsin expression specifically to dopamine projection neurons. Specifically, we injected a retrograde canine adenovirus expressing a cre-recombinase-dependent-Flp-recombinase (CAV-FLEX-flp, Fig. 4b) into the core or mShell of TH-cre rats followed by injection of AAVs expressing Flp dependent ArchT3.0 (AAV-fDIO-ArchT-eYFP) into the VTA. This produced expression of ArchT3.0 specifically in dopamine cells projecting the NAc core or mShell (Supplementary Fig. 6) allowing for selective inhibition of these distinct cell populations during extinction. We found that inhibition of mShell, but not core, projecting dopamine neurons reduced the consolidation/long-term retention of extinction memories (Fig. 4c). This demonstrates distinct VTA-dopamine cell populations projecting to core and mShell and reveals a novel function for the mShell projecting cells in modulating the persistence of extinction memories.

## Discussion

These findings show that activation of VTA-dopamine neurons during the expected shock omission time period is necessary for normal extinction learning and the upregulation of extinction-related plasticity markers in the vmPFC and amygdala. Notably, inhibition of VTA-dopamine neurons during the shock period of fear conditioning facilitates learning, suggesting that activity in VTA-dopamine neurons is not simply important for learning in response to any salient event. These results also reveal that distinct populations of VTA-dopamine neurons project to NAc core vs. mShell and that the mShell projecting cells are important for the formation of stable, long-term extinction memories. In contrast, a different VTA-dopamine cell population which projects to the vmPFC actively suppresses extinction as inhibition of their inputs to vmPFC during the shock omission period enhanced the retention of extinction memories. Together these findings suggest that activity in the VTA-dopamine system detects the omission of anticipated shocks and triggers a transition from fear to safety and that this is at least partially mediated through dopamine projections to the NAc.

One potential alternative interpretation of these results is that inhibition of VTA-dopamine neurons alone produced an aversive state that sustained fear responding during extinction[43,44]. However, inhibiting VTA-dopamine neurons during most of the auditory CS period had no effect on fear expression or extinction (Fig. 1d). Furthermore, pairing an auditory CS with VTA-dopamine neurons inhibition in place of shock was not sufficient to produce fear learning (Supplementary Fig. 3b, c). In addition, inhibiting VTA-dopamine cells during the CS period of fear

learning did not enhance fear conditioning (Fig. 2c). Finally, omission of expected aversive events increases dopamine neural activity and NAc dopamine release[7,10]. Together, this provides strong support for the conclusion that inhibition of this cell population specifically during the shock omission period of fear learning reduced extinction by blocking dopamine release evoked by unexpected shock omission and not because the inhibition by itself produced an aversive state.

These results reveal a novel function for the NAc projecting dopaminergic reward pathway and demonstrates a behavioral role for the enhanced dopamine release in the NAc mShell that occurs when an expected aversive outcome does not occur[9,10]. Specifically, our results show that activity in DA cells specifically during expected shock omission facilitates a switch from fear responding to extinction and that this is mediated partially through projections to the NAc mShell. This projection could work in concert with amygdala inputs to the NAc which also participate in extinction learning[45]. Surprisingly, our findings also show that activity during shock omission in mPFC projecting DA neurons actively opposes extinction learning while dopamine inputs to the amygdala during this time period do not appear to be involved in this process. Although vmPFC itself is important for extinction learning[1,2], our findings suggest that dopamine inputs to the vmPFC can play an opposing role. This is consistent with previous reports showing that D1 receptor activation in mPFC is important for reinstatement of fear memories following extinction[46] and that activation of mPFC projecting DA neurons produces aversive learning[34]. Possibly inconsistent with our results are reports that pharmacological blockade of dopamine receptors in both amygdala and vmPFC reduce extinction learning or consolidation of extinction[20,21], respectively. This discrepancy may be due to differences in the temporal specificity of dopamine manipulations as pharmacological approaches affect tonic dopamine signaling occurring during and after extinction while our optogenetic manipulations more specifically targeted responses occurring during the shock omission time period. In fact, phasic dopamine signaling in vmPFC, as may occur during shock omission, could more readily recruit D1 receptors which are less sensitive to dopamine than D2 receptors[47]. In contrast, D2 receptors could participate more in tonic dopamine signaling through their higher dopamine affinity and serve to facilitate fear extinction, a possibility directly supported by the finding that pharmacological D2 receptor blockade in vmPFC reduces extinction[21]. The different receptors may modulate distinct cell populations to produce these differential effects as D1 and D2 receptors are known to be preferentially expressed in distinct neuronal subpopulations in other brain regions[48,49].

A notable aspect of our findings is the discovery of a specific dopamine projection to the NAc mShell which is distinct from

**Fig. 3** Projection specific effects of dopamine nerve terminal inactivation on extinction. **a** Representative image showing a voltammetry optrode (carbon fiber probe (blue/gray) optical fiber (gold)) recording site (lesion denotes fiber tip) in the nucleus accumbens (NAc) and expression of ArchT in VTA-dopamine terminals in NAc coronal section. **b** Examples of dopamine release evoked by stimulation in medial forebrain bundle with (orange) and without (black) laser illumination in NAc of rats expressing ArchT-GFP (left) or GFP (right) in synaptic terminals of dopamine neurons in NAc (averages of five trials, laser on for 4 s, stim 2 s into laser period). **c** Optogenetic inhibition of VTA-dopamine terminals in NAc reduced evoked dopamine release (y-axis, % difference in [DA] in laser illumination trials relative to no-laser condition) in ArchT expressing animals, but not in GFP controls ($n = 8$ ArchT, 4 GFP, 5-trial average, *$p = 0.0263$, Student's $t$-test). **d, f, h** Paradigm for projection specific VTA-dopamine nerve terminal inhibition in NAc, mPFC and amygdala. Brain image adapted from ref. [70]. Copyright 1982, Elsevier. **e** Inhibition of VTA-dopamine terminals in NAc during shock omission period of extinction training does not affect extinction learning ($n = 9$ ArchT, 10 GFP, 2-trial average, $F_{10, 150} = 0.992$, $p = 0.4533$ no interaction, 2-way repeated measures ANOVA), but disrupted later retention (consolidation) of extinction memory (5-trial average, *$p = 0.0427$, Student's $t$-test). **g** Inhibition of dopamine terminals in vmPFC did not affect extinction learning ($n = 8$ ArchT, 9 GFP, 2-trial average, $F_{10, 130} = 1.230$, $p = 0.2778$ no interaction, 2-way repeated measures ANOVA), but enhanced consolidation of extinction memories (5-trial average, *$p = 0.0059$, Student's $t$-test). **i** Inhibition of VTA-dopamine terminals in lateral/basal amygdala (LA/B) during shock omission period of extinction training affects neither extinction learning ($n = 12$ ArchT, 14 GFP, 2-trial average, $F_{10, 240} = 0.831$, $p = 0.5990$ no interaction, 2-way repeated measures ANOVA), nor retention of extinction memory (5-trial average, *$p = 0.4043$, Student's $t$-test). Error bars indicate SEM. All (**a, d, f, h**) scale bars = 1 mm

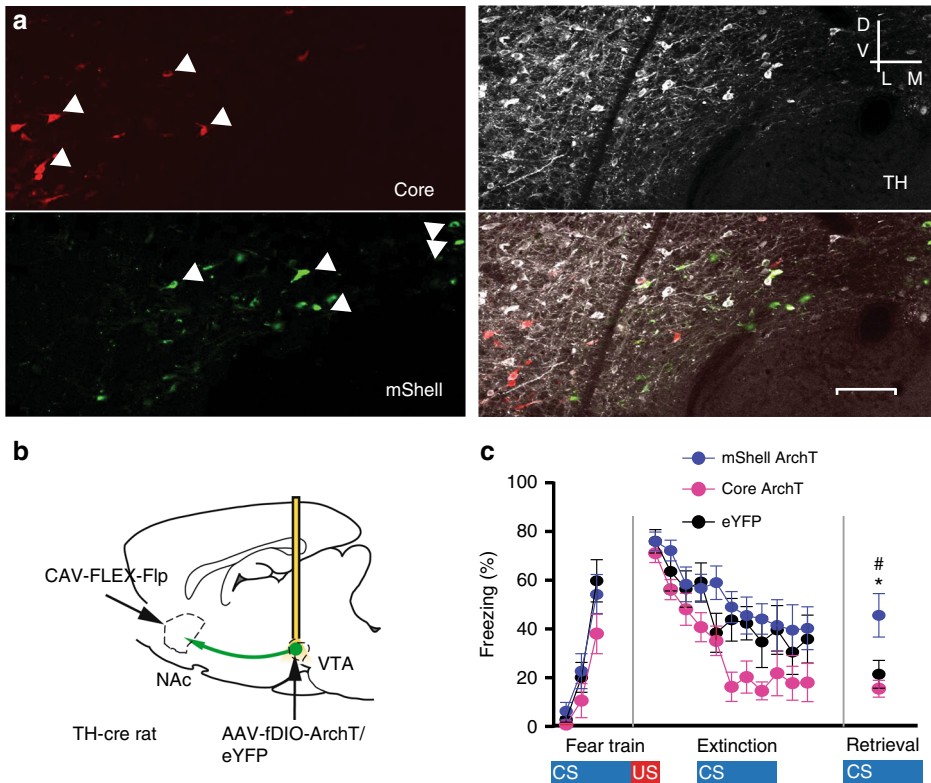

**Fig. 4** VTA-dopamine neurons projecting to NAc mShell facilitate consolidation of extinction learning. **a** Double retrograde tracer injections into NAc labeled unique populations of mShell (green, lower left) and core (red, upper left) projecting VTA-dopamine neurons (TH labeled, white, upper right). Overlay at bottom right. Arrowheads denote retrogradely labeled/TH + cells. **b** Schematic of optogenetic viral approach for inhibiting NAc mShell or core projecting VTA-dopamine neurons. Retrograde CAV-FLEX-flp (cre-dependent flp) virus injection into NAc mShell or core and AAV-fDIO-ArchT/eYFP or AAV-fDIO-eYFP (flp dependent) injections into VTA of TH-cre rat. Brain image adapted from ref. [70]. Copyright 1982, Elsevier. **c** Inhibition of mShell projecting VTA-dopamine neurons during shock omission period of extinction training reduced extinction memory retention. All groups extinguished equally ($F_{20, 230} = 1.19$, $p = 0.26$ interaction, two way repeated measures ANOVA), but freezing was significantly higher at the "retrieval" time-point in the mShell projecting group ($n = 9$, green) compared to both core projecting ($n = 8$, red) and eYFP ($n = 9$, black) groups ($F_{2, 24} = 5.620$, $p = 0.010$ one-way ANOVA, * and # denote significant differences between mShell group and both core and eYFP groups using Holm–Sidak posthoc tests). Error bars indicate SEM

projections to the NAc core and regulates fear extinction. This is consistent with a previous in vivo microdialysis study which found that dopamine in mShell increased at the cessation of an aversive predictive cue[10], while dopamine levels decreased in response to the cue itself in the NAc core[9,10]. The changes in dopamine levels in both regions were only present early in extinction learning, further suggesting an involvement in learning/consolidation. The enhanced dopamine signaling in mShell restricted to early extinction trials also suggests that dopamine release there reflects a prediction error signal that guides learning. The medial to lateral gradient of mShell vs. core projecting dopamine neurons also extends to more laterally situated VTA-dopamine populations which project to the lateral Shell (lShell)[50,51]. Interestingly, a recent study demonstrated that GABAergic D1 receptor expressing medium spiny neurons in NAc mShell and lShell have distinct connectivity with reciprocally connected and distinct mShell and lShell projecting VTA-dopamine neurons[51], suggesting a possible mechanism for individualized regulation of these different populations of dopamine neurons. While another microdialysis study[52] reported increases in dopamine in mShell in response to aversive predictive cues, because of the limited temporal specificity of the microdialysis technique this increase could reflect CS offset responses similar to what has been seen in previous work[10]. This may be related to the role of NAc dopamine in pain relief learning[11–13], which may

share similar neural substrates with fear extinction. Together with our results, these previous studies support a role for dopamine release in the NAc mShell in modulating fear extinction memory consolidation. The fact that inhibition of this dopamine projection only affected long-term retention of extinction, while the VTA-dopamine cell body manipulations affected both extinction learning and retention suggests that there may be other populations of dopamine neurons which project to different brain regions that participate in extinction learning. These populations may work together with substantia nigra dopamine projections to the dorsal striatum which have been implicated in fear extinction[53].

These results also suggest a differential involvement of NAc core vs. mShell in extinction learning, but how dopamine in mShell relates to MAPK phosphorylation in vmPFC and amygdala is not clear. One possibility is that the mShell regulates vmPFC and amygdala through basal ganglia connectivity loops. In fact, the NAc projects indirectly to both the amygdala and vmPFC[54–58]. Furthermore, previous work has demonstrated that the NAc core and mShell are functionally distinct[37,54,59] and maintain differential connectivity with the rest of the basal ganglia[54]. For example, anatomical studies have found that vmPFC projects preferentially to NAc mShell while dorsomedial PFC preferentially innervates NAc core, though these distinctions are not absolute[54–56]. NAc projects to ventral

pallidum, which projects in turn to distinct thalamic subregions which innervate either vm- or dorsal mPFC (dmPFC) as well as amygdala[54–58]. From the perspective of fear and reward learning these distinctions could be important as vm- and dmPFC have been implicated in distinct reward and aversive functions[2,60], with vmPFC participating in extinction and dmPFC being important in producing fear responses[2,61]. Providing functional support for the idea that NAc regulates extinction related networks, deep brain electrical stimulation of NAc facilitates fear extinction and MAPK phosphorylation in vmPFC and amygdala and this effect is dependent on neural activity in the NAc[62]. Furthermore, stimulation of amygdala inputs to NAc also enhanced extinction learning and activated vmPFC and, to a lesser extent, dmPFC neurons[45]. Together, this suggests that the NAc mShell modulates vmPFC and amygdala function during fear extinction learning, possibly through distributed, but preferential connectivity with these regions.

These findings could also have relevance for the treatment of human anxiety disorders. The importance of dopamine in fear extinction learning in humans was highlighted by one study reporting that a polymorphism in the dopamine transporter 1 (DAT1) gene, which likely increases dopamine levels, enhanced fear extinction[63]. Similarly, pharmacologically enhancing dopamine neurotransmission in humans facilitates fear extinction learning[64]. Exposure therapies such as extinction are first line treatments for anxiety disorders such as PTSD. Our results combined with the human work implicating the dopamine system in extinction suggest that enhancing dopamine signaling, and particularly dopamine release in the NAc during extinction could be an important therapeutic adjuvant when combined with exposure therapy. This could be accomplished through pharmacological means[64] and/or through counterconditioning[45,65,66] (but see ref. [67]), a treatment technique which combines exposure therapy with rewarding experiences and may preferentially engage the NAc projecting dopamine circuit.

## Methods

**Subjects**. Male Long-Evans TH-Cre rats[24] 9–20-week old were housed individually in temperature and humidity controlled environments on 12-h light-dark cycles (lights on from 8AM to 8PM) and provided with food and water ad libitum. TH-Cre breeding males were mated with 9 weeks old or older Long-Evans females (Japan SLC, Inc) in 2 week breeding periods to generate the TH-Cre colony. Littermates were assigned to experiment and control groups whenever possible. Experiments were all conducted during the animal's light cycle. All procedures conformed to protocols approved by the Animal Care and Use Committees of the RIKEN Brain Science Institute.

**Viruses**. AAV5-EF1α-DIO-eNpHR3.0-eYFP, AAV5-EF1α-DIO-eYFP, AAV9-CAG-FLEX-ArchT-GFP, AAV9-CAG-FLEX-GFP were produced by the University of North Carolina Vector Core (UNC, North Carolina, USA). CAV2-FLEX-flp was produced by the Montpellier vector core. AAV9-CAG-fDIO-eArchT3.0-eYFP and AAV9-CAG-fDIO-eYFP were produced and packaged in our lab.

**Virus or tracer injection and implantation**. Animals were induced and maintained under isoflurane anesthesia (Pfizer) at 1.5–3%. Animals were then placed in a stereotaxic device (Kopf Instruments) for virus injection and optical fiber insertion. For VTA manipulation experiments, AAV5 constructs were injected bilaterally into the VTA (AP -5.4, ML ±0.8, DV -7.9) using a Hamilton syringe (catalog #80100) attached to a Harvard Apparatus pump (PHD2000) at a rate of 0.1 µl/min (1 µl per side). 20 min post-injection, injection cannulae were withdrawn, and a dual fiber optic cannula (200 µm diameter, NA 0.37, Doric Lenses) was inserted bilaterally dorsal to the injection sites (AP -5.7, ML ±0.8, DV -7.6). Fiber optic cannulae were secured using head screws, a layer of Super Bond cement (Sun Medical) followed by application of acrylic dental cement. Animals were trained 3 weeks post-surgery. For terminal manipulation and voltammetry experiments, AAV9 constructs were injected bilaterally at 0.1 µl/min into four sites (0.5 µl per site) in VTA (AP -6.2 and -5.3, ML ±0.7, DV -8.4 and -7.4). For terminal manipulations, 6–9 weeks after the virus injection animals were anesthetized again and implanted with two optical fibers (200 µm diameter, NA 0.39, Thorlabs) on each side of the NAc (AP + 1.5, ML ±1.5, DV -6.75), LA (AP -2.9, ML ±5.3, DV -8), or IL (30° angle at AP + 2.8, ML ±3.1, DV -3.5; tip of the fibers at

AP + 2.8, ML ±0.9, DV -4.6). For manipulation of mShell or core projecting VTA-dopamine neuron experiments, CAV2-FLEX-flp was injected bilaterally at 0.1 µl/min (0.3 µl/site) in either NAc shell (AP + 1.0 and + 2.0, ML ±0.6, DV -7.6) or core (AP + 1 and + 2, ML ±2.0, DV -7.6) and flp-dependent AAV constructs (1.0 µl/side) were injected into VTA (AP -5.7, ML ±0.8, DV -7.9). A dual fiber optical cannula was implanted above the VTA using the same procedures described above. Animals were then allowed to recover for one week before behavioral experiments began at which time they were handled for 5 min each, 4 days before fear conditioning. For retrograde tracer experiment, 0.3 µl of Alexa Fluor 555 or 647 conjugated cholera toxin subunitB was ipsilaterally injected into Nac shell (AP + 1.5, ML 0.6, DV-7.6) and core (AP + 1.5, ML 2.0, DV-7.6).

**In vivo surgery and stimulation**. For voltammetry experiments, rats injected with AAV9 constructs in VTA 6–12 weeks previously were anesthetized with isoflurane and placed in a stereotaxic frame. Anesthesia was maintained with 0.5–1% isoflurane inhalation concentration. Craniotomies were made over the nucleus accumbens (1.9 mm anterior, 1 mm lateral relative to bregma) and medial forebrain bundle (-2.8 mm posterior, 1.8 mm lateral) in one hemisphere (left) for stimulating and recording electrode insertion. A third craniotomy was made over the opposite hemisphere for reference and auxiliary electrode insertion. A carbon fiber (cf)-optrode was lowered into the NAc (~7.2 mm V) with the cf-containing silica tube placed at the coordinates above. One silver wire (auxiliary electrode) and one silver/silver chloride wire (reference electrode) was inserted into the cortex of the contralateral hemisphere. A pair of stainless steel bipolar stimulating electrodes were lowered to the MFB. The dorsal–ventral placement of the stimulating electrode was adjusted to obtain maximal dopamine overflow during initial stimulation (24 pulses, 60 Hz, 0.3 mA, 4 ms/pulse). After stable and maximal striatal dopamine release was confirmed, trials with stimulation preceded by light application (Laser) alternated with trials without light application (No Laser).

**Behavioral conditioning experiments**. Animals were randomly assigned to groups prior to the start of each experiment. Animal numbers/groups were decided based on standard practices in fear conditioning and extinction experiments. During fear conditioning, animals were placed in a sound-isolating chamber (MED-PC, Med Associates) and presented with auditory conditioned stimuli (CS) paired with an aversive footshock unconditioned stimulus (US) 3X. The CS was a series of 20 kHz-pitch tone pips presented at 1 Hz (250 ms pulse width, 20 pulses, 74 dB, 750 ms inter-pip interval). The shock US (1 s, 0.7 mA) began at the onset of the last pip and lasted for 1 s (so outlasted final pip by 750 ms). Variable intertrial intervals (ITIs) averaged at 130 s were used. During fear extinction training 24 h later, animals were attached to a laser through fiber optic cabling and placed in a novel chamber with fluorescent lights off and a different scent in the box and the CS was presented repeatedly (22 ×) without the shock US. During each CS presentation laser illumination occurred starting either at the onset of the final pip and lasting for 3 s (the shock omission period) or throughout the auditory CS period (light onset occurred 400 ms prior to CS/first pip onset and terminated 50 ms after final pip termination). For the inhibition during shock omission experiments, the 3 s laser stimulation period was selected because a previous study[10] showed that dopamine levels increased in the NAc mShell after an expected shock was omitted and that the dopamine peaked after the offset of where the shock would have occurred. Intertrial intervals were variable averaging 150.9 s. Before each experiment laser power was adjusted to 12–15 mW of power at the tip of optical fibers. During extinction retrieval trials 24 h later, animals were returned to the extinction context with the same lighting and scent conditions as extinction training, and given five presentations of the same CS with variable ITIs averaging 150 s. Freezing levels were averaged in two trial bins during extinction training, and in a five trial average during testing. In the pMAPK study, control animals were placed in the same environment but without tones for the same amount of time. In the LA/B terminal inhibition studies, six of the ArchT and seven of the GFP injected animals were trained using a 1 kHz-pitch tone CS after previously being trained with 5 kHz-pitch tones. Twenty-four hours later, the animals were given extinction trials with light inhibition as in other animals, but with 1 kHz tones. The rest of the LA/B animals received the same training as the other experiments described above. No significant differences in freezing rates were found between groups experiencing 1 kHz vs. 5 kHz extinction and extinction retrieval trials, so their data were grouped. Freezing behavior during the 20 s CS period of fear training and during the 5 CS presentations of extinction retrieval were scored automatically using the Video Freeze program v2.7.1 (SOF 843, Med Associates) or a customized optical flow-based freezing program designed in Matlab. To avoid conflating freezing with sleeping during the longer extinction trials, all extinction freezing was scored manually by a rater blind to the identity of the treatment groups using a stopwatch. Freezing was defined as the cessation of all bodily movements except for breathing or sleep. The methods used to quantify freezing were identical across all experimental runs and were the same across all experimental groups.

For fear conditioning studies (Fig. 2) stimuli and presentation were identical to that described above for fear conditioning before extinction except that laser onset occurred 400 ms prior to shock US onset and was turned off 50 ms after US offset (total duration = 1.45 s). For the offset control group, laser onset occurred 30–50 s (pseudorandom selection) after US offset. 24 h later animals were presented with 5 tone CSs as described above during a retrieval test.

**Histology**. Within 3 days of the final experiment, animals were overdosed with 25% chloral hydrate and perfused with paraformaldehyde (4% in PBS). Brains were then post-fixed in a 30% sucrose PBS solution and then embedded in optimal cutting temperature (OCT) compound (Sakura Finetek) and sliced into 40 μm coronal sections using cryostat. Slices were then mounted on subbed glass slides (Matsunami, FRC-02), coverslipped using ProLong Gold antifade reagent (Molecular Probes) for microscopic examination. Additional sections were stored in 0.05% Sodium Azide PBS. In the pMAPK study, animals are sacrificed and perfused one hour after extinction training or one hour after being placed in the box for the same amount of time. Animals were excluded from analyses if virus expression was minimal/off-target or if fiber optic placements were incorrect (assessed blindly).

**Immunohistochemistry**. For verification of TH specificity of expression and visualization of terminals, immunohistochemistry was performed prior to mounting/coverslipping. Sections were washed 3 × in PBST (0.3% Triton-X in PBS) and blocked for 30 min in 2% bovine serum albumin (BSA) in PBST. Slices were then incubated in primary antibodies diluted in BSA-PBST overnight at 4°C in the dark. Next day, sections were washed 3× in PBS and incubated for 1 h in secondary antibodies (if necessary) diluted in BSA-PBST. After rinsing with PBS, slices were mounted as stated above. Primary antibodies used were goat anti-GFP Alexa Fluor 488 conjugate (1:1000, sc-5385 Santa Cruz), rabbit or mouse anti-TH (1:2000, AB152 MAB5280 Millipore), rabbit or mouse anti-GFP (1:2000, A11122 A11120 Life Technologies). Secondary antibodies used were goat anti-rabbit or goat anti-mouse Alexa Fluor 488 or 594 (1:500, A11034 A11029 A10037 A11032 Invitrogen). In the pMAPK study, sections were incubated in 1% hydrogen peroxide in PBS for 30 min before wash. The primary antibody used was rabbit anti-phospho-p44/42 MAPK Erk1/2 Thr202/204 (1:2000, #9101 Cell Signaling Technology). Sections were incubated for 72 h at 4°C. After incubation and 3 × PBS wash, sections were placed in biotinylated goat anti-rabbit IgG for 30 min (1:200, Vectastain Elite PK-6101 ABC HRP Kit, Vector). Following another 3 × PBS wash, sections were incubated in avidin-biotinylated HRP for 30 min (ABC HRP Kit) and washed again 3 × PBS. For visualization, slices were placed in a solution containing diaminobenzidine (DAB), hydrogen peroxide, and nickel (DAB Substrate Kit, SK-4100 Vector) for 5–10 min until staining of cell bodies was apparent. We used the same staining time for sections from all groups once the sufficient amount of time was determined. Sections were washed in distilled water, mounted on gelatin-coated slides, dehydrated using a series of 70%–95%–100% EtOH and finally xylene solutions, and coverslipped in Permount (Fisher).

**Electrochemistry**. Electrochemical recordings using FSCV were made in anesthetized rats with carbon fiber (cf) microelectrodes. The cf electrodes consisted of 7-μm-diameter carbon fibers (Goodfellow, Cambridge, England) threaded through silica tubing (100 μm i.d., 160 μm o.d., Polymicro Technologies) backfilled with epoxy resin. To make a cf optrode, the cf-threaded silica tubing was appended with silver paint (Dottite, Fujikura Kasei, Japan) to the metal holder for a 200 μm-diameter optic fiber such that the tubing extended parallel with the optic fiber at a distance of approximately 0.5 mm. The cf tip was cut to extend approximately 300 μm from the tip of the silica tubing. A commercial counter electrode-grounded type potentiostat (Model HECS-972E, Huso Electrochemical Systems, Kawasaki, Japan) was used for electrochemical recordings. Data acquisition was performed by a commercial control/recording system (TH-1; ESA Biosciences, Inc., MA, USA with two multifunction boards (NI-PCI-6221, National Instruments, TX, USA) implemented on a Windows PC (MODEL). The gain of the amplifier was 500 nA/V and the low-pass filter time-constant was 0.2 ms. Voltage scans from -0.4 V to 1.3 V vs. Ag/AgCl reference, and back from 1.3 V to -0.4 V, were applied to the cf electrode. This triangle-positive waveform was repeated at 10 Hz. Using this waveform, in vitro calibration of the cf electrodes was performed where a flow of phosphate-buffered saline (PBS) was switched to PBS containing known concentrations of dopamine, pH shifts or adenosine. The pH shifts were set using a pH-meter and adjusted with HCl or NaOH solution. Further calibration of light-generated artifact signals was made in vivo (see Electrochemistry analysis).

**Confocal imaging**. For quantification of the specificity of TH expression (Supplementary Fig. 1), VTA sections were doubled stained for TH (594 mm) and GFP (488 mm) and imaged under a confocal laser microscope (Fluoview FV-1000, Olympus). Images of lateral VTA were taken with a 20× objective. Counts of green cell bodies (GFP positive cells) which were also labeled for TH (red) were made and averaged across different sections as percentage of double labeled GFP+ cells. Assessment was performed by an individual blind as to treatment condition.

**Data analysis**. Two-way ANOVA, two-way repeated measures ANOVA, and unpaired Student's t-tests were calculated using Prism 6.0 (GraphPad). Significant interactions or main effects in the ANOVAs were analyzed for multiple comparisons ($\alpha = 0.05$) using Holm–Sidak test, which gives greater power than the Bonferroni and Tukey methods[68]. In the pMAPK study, immunoreactivity was quantified using four slices for each animal imaged with 10 × tiles on a microscope in bright field (BX63, Olympus). An experimenter blind to the identity of the groups cut out the IL and LA areas in the images and thresholded the brightness levels to obtain a count of the number of cells in the predefined areas using ImageJ. Results were calibrated by dividing by the area where counts were made to get counts of number of positive cells per $mm^2$. All behavioral experiments were run at least 2 × with groups consisting of experimental and control animals.

**Electrochemistry analysis**. Voltammetric data were first analyzed as background-subtracted currents relative to the mean of 10 waveform applications 1 s before the time of light application (3 s before electrical stimulation). The differential currents across the voltage scan were analyzed with principal component regression analysis using modified TH-1 CV-Analysis software (Paul Phillips Laboratory). Calibration templates were constructed using in vivo dopamine and pH templates that corresponded to known values obtained in vitro. In vivo adenosine templates were additionally used but were assigned nA current values for regression instead of concentration. Although optic fibers were not directly oriented toward the cf electrodes, 600 nm wavelength light application generated a voltammetric artifact similar to that observed previously with 460 nm[69] (Supplementary Fig. 4a and c). Although the same laser power was applied to all optic fibers, the amplitudes of light-generated artifact were unique for each animal due to variation in relative tip position between the optic fiber and cf probes. Therefore, calibration curves for light-generated current voltage curves were constructed based on graded laser power within each animal. Application of principal component regression analysis to calibration templates that included these light-generated calibration curves successfully separated DA-like signals from light-generated signals. The DA-calibrated signals were averaged over five stimulation trials of Laser and No Laser within each subject. Since there was no significant difference in dopamine release evoked in ArchT vs. GFP animals in No Laser trials (Supplementary Fig. 5) we calculated the percent difference in the peak value during light application relative to without light for each group and compared their means using an unpaired t-test.

**Code availability**. The code that support the findings of this study are available from the corresponding author upon reasonable request.

**Data availability**. The data that support the findings of this study are available from the corresponding author upon reasonable request.

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

## Acknowledgements

The authors would like to thank Lindsay Laurie, Jennifer Takahashi, Noah Miller and Yanqiu Tao for technical assistance. This work was supported by KAKENHI (15H04264, 25116531, 16H05928), the Strategic Research Program for Brain Sciences from the Ministry of Education, Culture, Sports, Science and Technology (11041047) and Kao Corporation funding to J.P.J. as well as KAKENHI (25871125) to R.L.

## Author contributions

R.L., A.U., and J.P.J. designed the experiments and wrote the manuscript. A.W. and T.J. M. designed and carried out the voltammetry experiments. R.L., A.U., A.W., L.A., and J. K. carried out the experiments. R.L., A.U., A.W., L.A., and J.K. analyzed the results.

## Additional information

**Competing interests:** The authors declare no competing interests.

