## [Peer Review File · Nature Communications]

Reviewers' comments:

Reviewer #1 (Remarks to the Author):

The experiments presented in the manuscript combine sophisticated tools to study a question of fundamental importance. However, I have two major concerns regarding the design of the experiments in relation to the question, and their internal consistency. These concerns are sufficiently worrying that I recommend rejection of the manuscript in its present form.

Major Concern 1:

In conditioning, the 1 s US co-terminated with the last pip of the CS. In the target condition, stimulation started at the onset of the final pip and lasted for 3 s [2000 ms post CS]. In the control for stimulation per se, it started 400 ms prior to CS onset and terminated 50 ms after CS termination. Given that the stimulation bracketed the period of the expected US in both cases, the data collected in the manuscript implies that there is a critical period within 50-2000 ms after the CS in which the absence of expected shock is registered by dopamine neurons in the VTA. This seems reasonable, but it is not the same as saying that dopamine neurons in the VTA code for the absence of expected shock. In fact, the parameters used for conditioning in these experiments are not conducive to addressing this question. Specifically, as the US co-terminated with the CS in conditioning, there is never an explicit moment (or set of moments) in which the animal will have expected the US to occur and this expectation was violated. All the animals could do in these experiments was use the offset of the CS to infer that the shock did not occur when it should have, and as such, it is difficult to see how dopamine neurons in the VTA could have coded for the absence of expected shock AT THE TIME WHEN SHOCK WAS IN FACT ABSENT.

Major Concern 2:

In the abstract, the authors state: "We report that when an expected aversive outcome does not occur, activity in a subpopulation of midbrain dopamine neurons is necessary to extinguish behavioral fear responses and engage molecular signaling events in extinction learning circuits." This particular claim is misleading: it suggests a demonstration that the midbrain dopamine neurons that are critical for extinction (i.e., those projecting to NAc) also trigger a cascade of events in prefrontal (IL) and amygdala (LA) regions that are critical for extinction. However, this is not shown in the manuscript. In fact, the two experiments reported in the manuscript appear to be quite distinct because they are not linked anatomically: i.e., the shift from a focus on the LA/IL to the NAc is surprising. Clearly, the manuscript lacks an experiment bringing together the experiments shown in Figure 1 and those in Figure 2.

The following is also stated in the abstract/summary: "This demonstrates a novel function for the canonical dopamine reward system and reveals distinct behavioural roles for different dopamine neuron projections." In light of the data reported, this appears to be a massive overstatement. What distinct behavioural roles are being shown here?

Other minor points:

1. Controls that receive stimulation during the CS deal with concerns re stimulation per se. The authors also show that stimulation of VTA dopamine neurons is not sufficient to produce de novo conditioning of a stimulus. However, a better test of this as an explanation of the data in extinction would be to show that stimulation of these VTA neurons does not produce conditioning of one stimulus subsequent to the actual conditioning of another. Alternatively, it would be useful to know whether stimulation of these VTA neurons affects reconditioning of the same stimulus.

2. Why should these effects be specific to extinction? Is there anything to suggest that they would

NOT be observed in acquisition?

E.g., If dopamine neurons in the VTA code for the affective state of relief that accompanies extinction of fear, one might expect opposite effects of the manipulation on acquisition. Alternatively, if VTA dopamine neurons simply code for the expectation of some event or its absence, one might expect a similar effect of the manipulation on acquisition.

3. Re experiments shown in Figure 1: It would have been appreciated if some Ephys data had been included to demonstrate that DA neurons were actually silenced.

4. Re experiments shown in Figure 2: Why the switch from NpHR to ArchT across experiments? And why no FSCV for LA and IL?

5. The discussion leans very heavily on the claim that manipulation of dopamine neurons in the VTA influences extinction retention via regulation of dopamine release in the NAc. This component of the story clearly emerged as an afterthought once the effects of VTA inhibition on dopamine release in the IL and BLA didn't turn out as expected. It creates something of a tension within the paper (along the lines indicated in Major Concern 2). Even if this part of the story is considered in isolation from Experiment 1, the data in relation to the role of the VTA-NAc projection in extinction is not persuasive (see Figure 2e). What were the baseline levels of freezing in the retention test?

6. The method section lists three distinct approaches to measuring freezing behaviour. This is quite confusing. Were distinct methods used across the different experiments? If yes, this is a bit of a concern.

7. Given the distinct roles played by subregions (IL vs. PL) of the mPFC during extinction, I'd suggest that the authors use the term IL-mPFC rather than mPFC alone.

8. There is a need to clarify the n for the experiments shown in Figure 2.

Reviewer #2 (Remarks to the Author):

In this report, Luo and colleagues explore the contribution of VTA neurons and their projection to the forebrain in the extinction of conditioned fear in rats. Optogenetic inhibition of VTA neurons in TH-cre rats produced an impairment in extinction learning and retrieval when light was applied to the VTA during the time of the expected US after the CS, but not during the CS period. The extinction retrieval impairment was reproduced by optogenetic inhibition of VTA terminals in the nucleus accumbens (but not the amygdala); interestingly inhibition of VTA terminals in the mPFC facilitated extinction retrieval. In no case, did terminal inhibition affect freezing during the extinction session, which contrasts with the results obtained after VTA inhibition. Lastly, the authors produce evidence that VTA inhibition reduces electrically-evoked dopamine release in the nucleus accumbens, although this was a modest reduction in release. Overall, this is an interesting study that implicates dopaminergic transmission in signaling outcome prediction errors--in this case the omission of an expected shock US--during fear extinction. The work is technically sound, the analyses are appropriate, and the conclusions are supported by the results. This topic is exciting and timely and will be of broad interest to the readership.

There are, however, some issues for the authors to consider: 1) DA release in the NAc is only modestly inhibited by optogenetic inhibition of VTA terminals in NAc. One would expect that a more

robust inhibition of DA would be required to influence extinction retention. As it stands, it appears that substantial extinction does in fact occur with both VTA inhibition and VTA terminal inhibition in NAc. A no-extinction control group would help to index the degree to which extinction was suppressed in the experimental groups. 2) Previous voltammetry recordings of DA release in NAc during fear extinction (Badrinarayan et al) showed opposite changes in the NAc core and shell (decreases in core during the CS; increases in shell after CS), and revealed that these changes occurred during the earliest extinction trials. The terminal illumination in NAc would likely have limited release at both core and shell terminals--how would shutting off increases in shell and decreases in core account for the observed behavioral effect? 3) Given that prediction errors are highest during the earliest extinction trials (which is reflected in DA dynamics in NAc during extinction), one would predict that opto inhibition of VTA would impair extinction when delivered during early extinction trials, but not later trials. This would be an important piece of evidence to show that the VTA manipulation is most effective when prediction error is greatest, but ineffective after extinction learning has occurred. 4) Would extinction learning/retrieval occur during paired trials during the "extinction" session if the VTA is activated during US delivery? Stephen Maren, TAMU

Reviewer #3 (Remarks to the Author):

This timely study addresses the role of the ventral tegmental area (VTA) dopamine (DA) system in fear extinction learning. To achieve this, the authors optogenetically silenced VTA neurons, as well as VTA projections to nucleus accumbens (NAcc), infralimbic cortex (IL), and basolateral amygdala (BLA) during extinction of auditory fear conditioning. Importantly, they compare silencing throughout the conditioned tone (no effect) with silencing limited to the time when shock would have occurred (blocked extinction), directly implicating dopamine neurons in the error signal driving extinction learning. While there has been plenty of theorizing about learning signals in extinction, this is the first study to directly test the hypothesis in DA neurons. Furthermore, they show that it is the projection from VTA to NAcc that mediates this effect.

The paper is well written, the effects are clearly presented in the figures, and it provides a fresh perspective on extinction learning. It is concise and accessible within the brief format of this journal. However, I do have the following concerns.

Major

1. Their hypothesis is that VTA outputs communicate the reward prediction error that drives extinction. Thus, blocking these projections should impair both extinction learning (within-session) as well as subsequent recall, as demonstrated in Fig 1C. The problem is that inactivation of the VTA-NAcc projection only impaired recall of extinction, but had no effect on extinction learning (see Fig 2E). Furthermore, prior work has suggested BLA mediates within-session extinction, but silencing the VTA-BLA pathways had no effect here (Fig 2I). Thus, this is not consistent with the hypothesis as stated. The authors might consider separating out rats based on VTA projections to the core vs shell of NAcc: perhaps revealing an effect on within-session extinction. If this does not work, then they should conclude that the VTA-NAcc projection mediates consolidation or recall of extinction memory, rather than extinction learning. This would be very similar to infralimbic cortex. Given the robust effect on within-session extinction shown in Fig 1C, this suggests that some other target of VTA may mediate extinction learning.

Minor

1) In the Discussion, the authors suggest that their IL findings are inconsistent with previous pharmacological studies due to a lack of temporal resolution. However, another possibility is the that

IL effects could be attributed to the different D1 vs D2 receptor locations within IL (Bourdy & Barrot 2012, Gangarossa et al 2013, Xu & Zhang 2015). The two receptors show different affinity for DA (D2 is 10x D1). Because of this, tonic DA release (e.g. during tone) will have an inhibitory effect by predominantly activating D2, whereas phasic DA release (e.g. at shock time) will have effects associated with D1 receptor activation (due to larger D1 presence than D2) (Dreyer et al 2010). The discussion could try to incorporate this information regarding DA's effect on IL.

2) In supplementary fig.4B, it is not clear what the colored lines represent (graph legend?).

3) Why did they switch from Halo in Exp 1, to Arch in Exp 2? This should be clearly stated.

4) Line 156 should cite Correia, S.S., McGrath, A.G., Lee, A., Graybiel, A.M. & Goosens, K.A. Amygdala-ventral striatum circuit activation decreases long-term fear. *eLife* 5(2016).

5) Typos: Line 97 "reinforce": "Reinforcer"; Line 97 "demonstrate": "demonstrates";

Bourdy R, Barrot M. 2012. A new control center for dopaminergic systems: pulling the VTA by the tail. *Trends Neurosci* 35: 681-90

Dreyer JK, Herrik KF, Berg RW, Hounsgaard JD. 2010. Influence of phasic and tonic dopamine release on receptor activation. *J Neurosci* 30: 14273-83

Gangarossa G, Espallergues J, Mailly P, De Bundel D, de Kerchove d'Exaerde A, et al. 2013. Spatial distribution of D1R- and D2R-expressing medium-sized spiny neurons differs along the rostro-caudal axis of the mouse dorsal striatum. *Front Neural Circuits* 7: 124

Xu L, Zhang XH. 2015. Distribution of D1 and D2-dopamine receptors in calcium-binding-protein expressing interneurons in rat anterior cingulate cortex. *Sheng Li Xue Bao* 67: 163-72

We thank the Reviewers for their insightful comments on our manuscript. While all Reviewers expressed some enthusiasm, they raised many substantive concerns. We have now added a great deal of new work to address these concerns and, in the process, significantly extended the scientific advance of the findings. Point-by-point responses to individual Reviewer comments are offered below (Reviewer comments are italicized, responses follow Reviewer comments) and major changes are highlighted in the text of the paper. Here we would like to first emphasize key additional experiments that were conducted and how these studies extend the manuscript.

- New anatomical experiments revealing distinct populations of VTA dopamine neurons that project to nucleus accumbens (NAc) core or medial shell (Fig. 4a)
- Adapted a retrograde virus/transgenic approach to express opsins specifically in core or shell projecting VTA dopamine neurons in rat (Supplementary Fig. 6)
- New optogenetic/behavioral experiments showing that the NAc medial shell projecting dopamine neurons, and not those which project to core, regulate extinction memory formation (Fig. 4b-c)
- New optogenetic/behavioral experiments showing that inhibition of VTA-dopamine neurons during the shock period, but not the auditory CS period, of fear conditioning enhances learning (Fig. 2)

The new experiments on the core vs. medial shell projecting VTA neurons are the first to demonstrate that completely separate populations of dopamine neurons innervate these two adjacent NAc regions. Furthermore, this new study also reveals a specific function for the medial shell projecting cell population in regulating fear extinction learning. This adds to the emerging picture of heterogeneity in the VTA dopamine system based on projection target and reveals a novel behavioral learning function of these medial shell projecting cells. The addition of the fear conditioning experiments also significantly extends the manuscript. Previous studies have suggested that dopamine is important in fear conditioning, but no studies have tested whether activation of dopamine neurons during specific temporal epochs of fear conditioning is important in fear learning. The new studies directly tested this question and found that inhibiting VTA dopamine neurons during the shock, but not during the auditory CS period of conditioning enhanced fear learning. This shows that shock evoked activity in these cells serves to restrain fear learning and suggests that VTA dopamine neurons are not simply important for encoding novel stimuli/events during learning.

Reviewer #1 (Remarks to the Author):

The experiments presented in the manuscript combine sophisticated tools to study a question of fundamental importance. However, I have two major concerns regarding the design of the experiments in relation to the question, and their internal consistency. These concerns are sufficiently worrying that I recommend rejection of the manuscript in its present form.

Major Concern 1:

In conditioning, the 1 s US co-terminated with the last pip of the CS. In the target condition, stimulation started at the onset of the final pip and lasted for 3 s [2000 ms post CS]. In the control for stimulation per se, it started 400 ms prior to CS onset and terminated 50 ms after CS termination. Given that the stimulation bracketed the period of the expected US in both cases, the data collected in the manuscript implies that there is a critical period within 50-2000 ms after the CS in which the absence of expected shock is registered by dopamine neurons in the VTA. This seems reasonable, but it is not the same as saying that dopamine neurons in the VTA code for the absence of expected shock. In fact, the parameters used for conditioning in these experiments are not conducive to addressing this question. Specifically, as the US co-terminated with the CS in conditioning, there is never an explicit moment (or set of moments) in which the animal will have expected the US to occur and this expectation was violated. All the animals could do in these experiments was use the offset of the CS to infer that the shock did not occur when it should have, and as such, it is difficult to see how dopamine neurons in the VTA could have coded for the absence of expected shock AT THE TIME WHEN SHOCK WAS IN FACT ABSENT.

Response: This is a very important point the Reviewer makes here and we understand their concern. If the shock coterminated with the end of the final pip and the CS inhibition, which terminated at the end of the last pip, didn't produce an effect on extinction learning, while the inhibition during the time period of shock omission did it would be a bit confusing. In fact the shock did not co-terminate with the final pip, but instead started at the onset of the final pip and lasted for 1 second. We mistakenly wrote this in the Methods section of the previous draft. We apologize for the oversight which has been corrected in the revised manuscript (Pg. 9).

Because the pips lasted 250 ms this means that the omission of shock period has a detection window of 750 ms beyond the pip as the shock starts at the last pip onset and lasts for 1 sec. We kept the laser on for a total of 3 seconds because a previous fear conditioning study examining the dynamics of dopamine release in the NAc medial shell when expected aversive shocks were omitted reported that dopamine release outlasted and in fact peaked after the explicit shock omission period¹ (see Fig. 3g-h from that paper). We now include a detailed discussion of this issue in the revised manuscript (Pg. 9) and thank the Reviewer for giving us the opportunity to correct this oversight.

Major Concern 2:

In the abstract, the authors state: "We report that when an expected aversive outcome does not occur, activity in a subpopulation of midbrain dopamine neurons is necessary to extinguish behavioral fear responses and engage molecular signaling events in extinction learning circuits." This particular claim is misleading: it suggests a demonstration that the midbrain dopamine neurons that are critical for

extinction (i.e., those projecting to NAc) also trigger a cascade of events in prefrontal (IL) and amygdala (LA) regions that are critical for extinction. However, this is not shown in the manuscript. In fact, the two experiments reported in the manuscript appear to be quite distinct because they are not linked anatomically: i.e., the shift from a focus on the LA/IL to the NAc is surprising. Clearly, the manuscript lacks an experiment bringing together the experiments shown in Figure 1 and those in Figure 2.

Response: We agree that an understanding of how the VTA dopamine-to-nucleus accumbens circuit interacts with amygdala and medial prefrontal cortex circuits during extinction is a very interesting question. In fact the basal ganglia maintains strong interconnections with the medial prefrontal cortex and amygdala²⁻⁶ and stimulation of amygdala inputs to the nucleus accumbens or NAc modulates neural activity in the mPFC and amygdala and enhances extinction learning^{7,8}. These findings provide a possible circuit link between the nucleus accumbens and traditional fear extinction circuitry. While we feel that a detailed exploration of this question is beyond the scope of the current study, we now include a more thorough discussion of this important future direction in the revised manuscript (pg. 7).

The following is also stated in the abstract/summary: "This demonstrates a novel function for the canonical dopamine reward system and reveals distinct behavioural roles for different dopamine neuron projections." In light of the data reported, this appears to be a massive overstatement. What distinct behavioural roles are being shown here?

Response: We can see the Reviewer's point. We were referring to the opposing effects on fear extinction with inhibition of NAc and mPFC terminal manipulations (reduced extinction with inhibition of NAc projecting dopamine neurons and enhanced extinction with inhibition of mPFC projecting dopamine cells). However, we acknowledge that we did not look at the role of these projections in *different behaviors* and instead showed that these projection have *opposing roles in the same behavior*. In the revised manuscript, we have removed references to distinct effects on behavior and focus on the opposing roles of these projections in extinction.

Other minor points:

1. Controls that receive stimulation during the CS deal with concerns re stimulation per se. The authors also show that stimulation of VTA dopamine neurons is not sufficient to produce de novo conditioning of a stimulus. However, a better test of this as an explanation of the data in extinction would be to show that stimulation of these VTA neurons does not produce conditioning of one stimulus subsequent to the actual conditioning of another. Alternatively, it would be useful to know whether stimulation of these VTA neurons affects reconditioning of the same stimulus.

Response: The Reviewer makes a good point here. While we feel we that the data in the original submission dealt with this fairly well, we acknowledge that this was not clearly articulated. We also now include new data that further supports our original interpretation of the data (see below). One thing we should clarify is that we did not perform any 'stimulation' (increasing neuronal activity) studies in our experiments. We only used optical inhibition of dopamine cells to test their necessity for the behaviors under study. The main issue here is whether this inhibition produced some sort of aversive state that substituted for the shock during extinction and thereby reduced extinction learning. As the Reviewer points out, if this were true then inhibition occurring during 19+ sec of CS should block extinction and it

does not (Fig 1d). This inhibition during the CS also had no effect on animals' behavioral freezing response to the CS. This shows that inhibition occurring during the CS was not sufficient to sustain fear responding during extinction or on the animals' cue-evoked freezing response. Furthermore, repeatedly pairing the CS with optical inhibition alone did not produce fear conditioning (Suppl. Fig. 3b-c), as noted by the Reviewer. This provides further support for the hypothesis that inhibition of dopamine cells alone was not sufficient to produce fear learning. Finally, we now **include new data** showing that inhibition of VTA-dopamine neurons during the auditory CS period of fear conditioning did not enhance fear learning (Fig. 2c). Combined, these experiments argue strongly against the interpretation that the inhibition of dopamine cells by itself provides some sort of aversive signal which blocks extinction. We now include a discussion of these important issues in the revised manuscript (pg. 3, 4 & 6).

2. Why should these effects be specific to extinction? Is there anything to suggest that they would NOT be observed in acquisition?

E.g., If dopamine neurons in the VTA code for the affective state of relief that accompanies extinction of fear, one might expect opposite effects of the manipulation on acquisition. Alternatively, if VTA dopamine neurons simply code for the expectation of some event or its absence, one might expect a similar effect of the manipulation on acquisition.

Response: This is a very interesting idea because, as the Reviewer points out, this experiment would allow us to better dissociate what these cells are doing in different forms of behavioral learning (fear and extinction). We have now added **additional fear conditioning experiments** which are included in the revised manuscript (pg. 4, Fig. 2). To briefly summarize, we see the opposite effect when we optogenetically inactivate VTA dopamine neurons during the shock period of fear conditioning (i.e. fear learning is enhanced). Furthermore, inactivation of VTA dopamine neurons during the tone CS period of fear conditioning has no effect on learning. Together, this extends the manuscript considerably and provides strong support for the idea that dopamine neurons are in fact encoding relief rather than simply the expectation of some event or its absence.

3. Re experiments shown in Figure 1: It would have been appreciated if some Ephys data had been included to demonstrate that DA neurons were actually silenced.

Response: We understand the Reviewer's concern here. We based our approach on previously published work^{9,10} showing that cell body inhibition using halorhodopsin suppressed dopamine release as well as dopamine neural activity (in-vitro and in-vivo)^{9,10}. We have now added a discussion of these papers to the revised manuscript (pg. 3).

4. Re experiments shown in Figure 2: Why the switch from NpHR to ArchT across experiments? And why no FSCV for LA and IL?

Response: We used ArchT in the terminal inhibition experiments because we did not see good terminal expression using NpHR and because previous work has shown the effectiveness of terminal inhibition using ArchT¹¹⁻¹³, particularly using short optical illumination periods similar to what we used¹¹. We

chose the NAc to confirm optogenetic control of dopamine release because it is a well-established site for dopamine measurement¹⁴⁻¹⁶. Also, among the neuromodulators detectable by FSCV, MFB stimulation elicits predominantly dopamine within the NAc¹⁷. Because FSCV does not readily differentiate dopamine from other monoamines and other release sites have not been well characterized it would be hard to interpret results from, for example, IL and LA. We have added a more thorough discussion of these issues in the revised manuscript (pg. 4).

5. The discussion leans very heavily on the claim that manipulation of dopamine neurons in the VTA influences extinction retention via regulation of dopamine release in the NAc. This component of the story clearly emerged as an afterthought once the effects of VTA inhibition on dopamine release in the IL and BLA didn't turn out as expected. It creates something of a tension within the paper (along the lines indicated in Major Concern 2). Even if this part of the story is considered in isolation from Experiment 1, the data in relation to the role of the VTA-NAc projection in extinction is not persuasive (see Figure 2e). What were the baseline levels of freezing in the retention test?

Response: While we do think that the effects of inhibition of VTA-dopamine projections to the NAc on extinction retention are clear, we understand the Reviewer's criticism here. We have now **added additional experiments** where we expand on the VTA-NAc projection in extinction and validate our original terminal manipulation experiments using an alternate approach. We've also reformatted the paper to focus more on this pathway and discuss how it can modulate plasticity in amygdala and mPFC through the known basal ganglia connectivity with vmPFC and amygdala (pg. 7). Specifically, we now show that there are separate populations of VTA-dopamine cells which project to the NAc core vs. medial shell and that the medial shell projecting, and not the core projecting, cells mediate the effect on extinction retention (Fig. 4). This fits nicely with previous work showing increases in dopamine release in medial shell when an expected shock is omitted¹ and links this functionally to a specific population of dopamine neurons. We have also **added the additional analysis** that the Reviewer requested on baseline freezing levels in the retention test and found no group differences (Suppl Fig. 5e). Elucidating this novel extinction circuit (and new functional dopamine pathway) adds considerably to the conceptual advance of the findings and we thank the Reviewer for giving us the opportunity to expand the paper in this way.

6. The method section lists three distinct approaches to measuring freezing behaviour. This is quite confusing. Were distinct methods used across the different experiments? If yes, this is a bit of a concern.

Response: We apologize for not making this clear. The same methods were used across all experiments. The differences in scoring come from the fact that we have different cameras/automated scoring systems for different boxes used for acquisition, testing and extinction. Furthermore, for within session extinction we always used hand scoring because we did not trust the automated systems to distinguish between sleeping and freezing (sleeping doesn't occur during fear conditioning or shorter retrieval tests). However, the methods were the same across all experimental groups and were identical across all experimental runs and would have no differential impact on the results. This is clarified in the revised manuscript (Pg. 10).

7. Given the distinct roles played by subregions (IL vs. PL) of the mPFC during extinction, I'd suggest that the authors use the term IL-mPFC rather than mPFC alone.

Response: We have changed this in the manuscript and now use vmPFC (and distinguish between IL/vmPFC and dorsal mPFC).

8. There is a need to clarify the n for the experiments shown in Figure 2.

Response: This has been clarified in the revised manuscript in (see Figure Legend for Fig. 3, formerly Fig. 2).

Reviewer #2 (Remarks to the Author):

In this report, Luo and colleagues explore the contribution of VTA neurons and their projection to the forebrain in the extinction of conditioned fear in rats. Optogenetic inhibition of VTA neurons in TH-cre rats produced an impairment in extinction learning and retrieval when light was applied to the VTA during the time of the expected US after the CS, but not during the CS period. The extinction retrieval impairment was reproduced by optogenetic inhibition of VTA terminals in the nucleus accumbens (but not the amygdala); interestingly inhibition of VTA terminals in the mPFC facilitated extinction retrieval. In no case, did terminal inhibition affect freezing during the extinction session, which contrasts with the results obtained after VTA inhibition. Lastly, the authors produce evidence that VTA inhibition reduces electrically-evoked dopamine release in the nucleus accumbens, although this was a modest reduction in release. Overall, this is an interesting study that implicates dopaminergic transmission in signaling outcome prediction errors--in this case the omission of an expected shock US--during fear extinction. The work is technically sound, the analyses are appropriate, and the conclusions are supported by the results. This topic is exciting and timely and will be of broad interest to the readership.

There are, however, some issues for the authors to consider: 1) DA release in the NAc is only modestly inhibited by optogenetic inhibition of VTA terminals in NAc. One would expect that a more robust inhibition of DA would be required to influence extinction retention. As it stands, it appears that substantial extinction does in fact occur with both VTA inhibition and VTA terminal inhibition in NAc. A no-extinction control group would help to index the degree to which extinction was suppressed in the experimental groups.

Response: This is a good point the Reviewer raises. For the Figure 1c data, we have now added **additional analyses** comparing the first tone CS of extinction with the final tone CS at the end of extinction and the first tone CS during the extinction Retention test. What we've found with is that in both dopamine cell body and terminal inhibition in NAc experiments there is some extinction occurring during the extinction training session (comparing the first tone CS presentation to the last, see Suppl. Fig. 3a and 5d). However, fear responses completely return (in the case of dopamine terminal inactivation in NAc) or partially recover (in the case of dopamine cell body inactivation) during the retention test occurring 24 hrs after training. One explanation for these subtle differences in the behavioral findings are differences in the effectiveness of cell body vs. terminal inhibition, as the Reviewer points out, though we saw similar effects with cell body inactivation of dopamine neurons

projecting to a specific subregion of NAc (see response to point 2 below). Thus another possibility is that other populations of dopamine cells in the VTA also participate in extinction learning and that together they participate in both within-session and promote long term retention of extinction. We have added these new analyses as well as a discussion of these important issues to the revised manuscript (pg. 3, 5 and 7).

2) Previous voltammetry recordings of DA release in NAc during fear extinction (Badrinarayan et al) showed opposite changes in the NAc core and shell (decreases in core during the CS; increases in shell after CS), and revealed that these changes occurred during the earliest extinction trials. The terminal illumination in NAc would likely have limited release at both core and shell terminals--how would shutting off increases in shell and decreases in core account for the observed behavioral effect?

Response: This is a very interesting idea the Reviewer raises here and we **conducted a number of new experiments** to test this question. In the Badrinarayan study they compared dopamine release in the core compared with medial shell, just adjacent to the core. Differences in the role of dopamine in the NAc medial shell vs core have not been studied as much as differences in lateral shell vs. core, likely because of the close proximity of medial shell and core. Furthermore, it was not clear whether distinct populations of VTA dopamine neurons projected to core vs. medial shell. In anatomical experiments we tested this and found distinct populations of VTA dopamine neurons which project to NAc core vs. medial shell (Fig. 4a). Next we examined whether optogenetic inactivation of either of these populations affected fear extinction. This was not trivial as the VTA sends both dopaminergic and GABAergic projections to the NAc and we did not feel it was possible to limit light spread to either structure in isolation using the terminal inhibition approach we'd used previously. To specifically label VTA-medial shell and VTA-core projecting dopamine neurons we adapted a retrograde virus/transgenic approach to express Archaelrhodopsin in the different populations of dopamine neurons in rat (Supplementary Fig. 6). Finally, we optogenetically inhibited these different cell populations during the shock omission period of extinction. We found that inhibition of the medial shell, but not core, projecting VTA-dopamine neurons reduced retention of extinction memories (Fig. 4b-c). Together these findings identify distinct populations of medial shell vs. core projecting VTA dopamine neurons and shows a specific effect on extinction of inhibiting the medial shell projecting cells, nicely complementing the previous study on dopamine release there during extinction¹. This adds considerably to the scientific advance of the work and we thank the Reviewer for suggesting these interesting experiments.

3) Given that prediction errors are highest during the earliest extinction trials (which is reflected in DA dynamics in NAc during extinction), one would predict that opto inhibition of VTA would impair extinction when delivered during early extinction trials, but not later trials. This would be an important piece of evidence to show that the VTA manipulation is most effective when prediction error is greatest, but ineffective after extinction learning has occurred.

Response: We have now added a discussion of the dynamics of the dopamine signal in the NAc medial shell and how this relates to our newly gathered data on the specific dopamine cells which project there (see pg. 7). However, we have already added a great deal of new work to this paper and feel that these additional experiments, while interesting, are not critical to our findings and beyond the scope of the current study.

4) Would extinction learning/retrieval occur during paired trials during the "extinction" session if the VTA is activated during US delivery? Stephen Maren, TAMU

Response: This is another interesting idea the Reviewer raises. We have now **collected new data** which speaks to this question. Specifically, we have now performed inhibition of dopamine cells during the shock US period of fear conditioning and found that this enhances fear learning. Based on this we anticipate that inhibition during overtraining as the Reviewer suggests would likely increase learning rather than causing extinction. We think this is because shock-evoked activity in dopamine neurons normally reduces fear learning and by optogenetically inhibiting this activity, learning is enhanced. A discussion of these issues can be found in the revised manuscript (pg. 4).

Reviewer #3 (Remarks to the Author):

This timely study addresses the role of the ventral tegmental area (VTA) dopamine (DA) system in fear extinction learning. To achieve this, the authors optogenetically silenced VTA neurons, as well as VTA projections to nucleus accumbens (NAcc), infralimbic cortex (IL), and basolateral amygdala (BLA) during extinction of auditory fear conditioning. Importantly, they compare silencing throughout the conditioned tone (no effect) with silencing limited to the time when shock would have occurred (blocked extinction), directly implicating dopamine neurons in the error signal driving extinction learning. While there has been plenty of theorizing about learning signals in extinction, this is the first study to directly test the hypothesis in DA neurons. Furthermore, they show that it is the projection from VTA to NAcc that mediates this effect.

The paper is well written, the effects are clearly presented in the figures, and it provides a fresh perspective on extinction learning. It is concise and accessible within the brief format of this journal. However, I do have the following concerns.

Major

1. Their hypothesis is that VTA outputs communicate the reward prediction error that drives extinction. Thus, blocking these projections should impair both extinction learning (within-session) as well as subsequent recall, as demonstrated in Fig 1C. The problem is that inactivation of the VTA-NAcc projection only impaired recall of extinction, but had no effect on extinction learning (see Fig 2E). Furthermore, prior work has suggested BLA mediates within-session extinction, but silencing the VTA-BLA pathways had no effect here (Fig 2I). Thus, this is not consistent with the hypothesis as stated. The authors might consider separating out rats based on VTA projections to the core vs shell of NAcc: perhaps revealing an effect on within-session extinction. If this does not work, then they should conclude that the VTA-NAcc projection mediates consolidation or recall of extinction memory, rather than extinction learning. This would be very similar to infralimbic cortex. Given the robust effect on within-session extinction shown in Fig 1C, this suggests that some other target of VTA may mediate extinction learning.

Response: This is an excellent point the Reviewer raises. We have **now conducted the additional experiments** that he/she suggests targeting the NAc core or medial shell dopamine projection. We found that separate populations of VTA dopamine neurons project to NAc core vs. medial shell (Fig. 4a). Importantly, inhibition of the medial shell, but not the core, projecting cells reduced the consolidation/retention of extinction (Fig. 4b-c). We did not see effects on within session extinction, similar to our results with inhibition of the dopamine terminals in NAc that we reported in the first version of the manuscript. Our conclusion from this is that dopamine neurons, as a population, are

important for extinction learning and memory and that the cells that project to medial shell participate in the consolidation process specifically. As the Reviewer suggests, we have now added a discussion of this to the manuscript with the conclusion being that there is another population of dopamine cells that we have not identified that contribute to the within session extinction effects we see with global VTA-dopamine inhibition (pg. 7).

Minor

1) *In the Discussion, the authors suggest that their IL findings are inconsistent with previous pharmacological studies due to a lack of temporal resolution. However, another possibility is that IL effects could be attributed to the different D1 vs D2 receptor locations within IL (Bourdy & Barrot 2012, Gangarossa et al 2013, Xu & Zhang 2015). The two receptors show different affinity for DA (D2 is 10x D1). Because of this, tonic DA release (e.g. during tone) will have an inhibitory effect by predominantly activating D2, whereas phasic DA release (e.g. at shock time) will have effects associated with D1 receptor activation (due to larger D1 presence than D2) (Dreyer et al 2010). The discussion could try to incorporate this information regarding DA's effect on IL.*

Response: This is an important point. We have now added a discussion of these issues with appropriate citations to the articles the Reviewer cites to the Discussion section of the revised manuscript (pg. 6).

2) *In supplementary fig.4B, it is not clear what the colored lines represent (graph legend?).*

Response: We apologize for not clearly stating this in the Figure Legend. The colors represent different calibration intensities (concentration for DA, change in pH value, laser output (%) for light artifact). Laser light was applied at graded intensities to obtain graded amplitudes for calibration. The different colors represent the different light intensities. This is now clarified in the figure legend for Suppl. Fig. 4.

3) *Why did they switch from Halo in Exp 1, to Arch in Exp 2? This should be clearly stated.*

Response: We used ArchT in the terminal inhibition experiments because we did not see good terminal expression using NpHR and previous work has demonstrated the effectiveness of terminal inhibition using ArchT^{11,12}. We now include a discussion of this in the revised manuscript (pg. 4).

4) *Line 156 should cite Correia, S.S., McGrath, A.G., Lee, A., Graybiel, A.M. & Goosens, K.A. Amygdala-ventral striatum circuit activation decreases long-term fear. eLife 5(2016).*

Response: This paper is now cited in this sentence (pg. 8).

5) *Typos: Line 97 "reinforce": "Reinforcer"; Line 97 "demonstrate": "demonstrates";*

Response: These typos are now corrected in the revised manuscript.

Bourdy R, Barrot M. 2012. A new control center for dopaminergic systems: pulling the VTA by the tail. Trends Neurosci 35: 681-90

Dreyer JK, Herrik KF, Berg RW, Hounsgaard JD. 2010. Influence of phasic and tonic dopamine release on receptor activation. J Neurosci 30: 14273-83

Gangarossa G, Espallergues J, Mailly P, De Bundel D, de Kerchove d'Exaerde A, et al. 2013. Spatial distribution of D1R- and D2R-expressing medium-sized spiny neurons differs along the rostro-caudal axis of the mouse dorsal striatum. *Front Neural Circuits* 7: 124

Xu L, Zhang XH. 2015. Distribution of D1 and D2-dopamine receptors in calcium-binding-protein expressing interneurons in rat anterior cingulate cortex. *Sheng Li Xue Bao* 67: 163-72

References

1. Badrinarayan, A. et al. Aversive stimuli differentially modulate real-time dopamine transmission dynamics within the nucleus accumbens core and shell. *J Neurosci* **32**, 15779-90 (2012).
2. Voorn, P., Vanderschuren, L.J., Groenewegen, H.J., Robbins, T.W. & Pennartz, C.M. Putting a spin on the dorsal-ventral divide of the striatum. *Trends Neurosci* **27**, 468-74 (2004).
3. Vertes, R.P. Differential projections of the infralimbic and prelimbic cortex in the rat. *Synapse* **51**, 32-58 (2004).
4. Berendse, H.W., Galis-de Graaf, Y. & Groenewegen, H.J. Topographical organization and relationship with ventral striatal compartments of prefrontal corticostriatal projections in the rat. *J Comp Neurol* **316**, 314-47 (1992).
5. Van der Werf, Y.D., Witter, M.P. & Groenewegen, H.J. The intralaminar and midline nuclei of the thalamus. Anatomical and functional evidence for participation in processes of arousal and awareness. *Brain Res Brain Res Rev* **39**, 107-40 (2002).
6. Groenewegen, H.J., Berendse, H.W., Wolters, J.G. & Lohman, A.H. The anatomical relationship of the prefrontal cortex with the striatopallidal system, the thalamus and the amygdala: evidence for a parallel organization. *Prog Brain Res* **85**, 95-116; discussion 116-8 (1990).
7. Correia, S.S., McGrath, A.G., Lee, A., Graybiel, A.M. & Goosens, K.A. Amygdala-ventral striatum circuit activation decreases long-term fear. *eLife* **5**(2016).
8. Rodriguez-Romaguera, J., Do Monte, F.H. & Quirk, G.J. Deep brain stimulation of the ventral striatum enhances extinction of conditioned fear. *Proc Natl Acad Sci U S A* **109**, 8764-9 (2012).
9. McCutcheon, J.E. et al. Optical suppression of drug-evoked phasic dopamine release. *Front Neural Circuits* **8**, 114 (2014).
10. Chaudhury, D. et al. Rapid regulation of depression-related behaviours by control of midbrain dopamine neurons. *Nature* **493**, 532-6 (2013).
11. Mahn, M., Prigge, M., Ron, S., Levy, R. & Yizhar, O. Biophysical constraints of optogenetic inhibition at presynaptic terminals. *Nat Neurosci* (2016).
12. Spellman, T. et al. Hippocampal-prefrontal input supports spatial encoding in working memory. *Nature* **522**, 309-14 (2015).
13. Ozawa, T. et al. A feedback neural circuit for calibrating aversive memory strength. *Nat Neurosci* **20**, 90-97 (2017).
14. Di Chiara, G. Nucleus accumbens shell and core dopamine: differential role in behavior and addiction. *Behav Brain Res* **137**, 75-114 (2002).
15. Rodeberg, N.T., Sandberg, S.G., Johnson, J.A., Phillips, P.E. & Wightman, R.M. Hitchhiker's Guide to Voltammetry: Acute and Chronic Electrodes for in Vivo Fast-Scan Cyclic Voltammetry. *ACS Chem Neurosci* **8**, 221-234 (2017).
16. Wassum, K.M. & Phillips, P.E. Probing the neurochemical correlates of motivation and decision making. *ACS Chem Neurosci* **6**, 11-3 (2015).
17. Park, J., Aragona, B.J., Kile, B.M., Carelli, R.M. & Wightman, R.M. In vivo voltammetric monitoring of catecholamine release in subterritories of the nucleus accumbens shell. *Neuroscience* **169**, 132-42 (2010).

Reviewers' comments:

Reviewer #1 (Remarks to the Author):

A dopaminergic switch for fear to safety transitions

Luo et al

This paper shows that inhibition of VTA dopamine neurons has contrasting effects on fear acquisition and extinction, identified the timing of these effects to the period when an expected US is not presented, and in the case of extinction, identified some of the circuitry through which these neurons exert their influence. The paper has been improved considerably since its original submission: the additional data (especially that collected in relation to fear acquisition) has helped to build a compelling story regarding the role of VTA dopamine neurons in fear learning and memory. It is suitable for publication in Nature Communications once some minor issues have been addressed in the paper.

In relation to the introduction: The final sentence of the first paragraph (excluding abstract) seems to exclude the possibility that the amygdala and/or vmPFC registers the absence of an expected aversive event. Not even the present findings exclude this (or these) possibilities. As such, some constraint should be applied when constructing arguments in this paragraph.

Re the analysis used to determine whether "extinction was completely blocked:" In Supp Fig 3A, it is claimed that, relative to the first trial of extinction, the level of freezing to the CS was lower at the end of extinction and in the retention test. One presumes that this data/analysis related to Group HpHR only – this should be clarified. While it is clear that freezing declined across extinction and was lower on the final trial relative to the first, it is not at all clear that the level of freezing on the first trial of the retention test was any lower than the first trial of extinction: the overlapping error bars for this comparison are a slight concern. I understand that this is a within-subject comparison, thereby rendering the error-bars unnecessary. Perhaps further justification could be given for selection of the Holm-Sidak post-hoc test that was used to support this statement? More generally, I don't understand what this analysis has added to the manuscript (at this point or anywhere else where it is repeated). It could be removed without harm to the overall story.

On page 3 (Lines 119+) it is stated: "In eYFP controls, extinction training increased pMAPK levels in lateral amygdala (LA) and infralimbic (IL) subregion of mPFC compared to unpaired controls." What was the unpaired control? The preceding text (and figure) describes a control group that is exposed to the context in the absence of the tone alone presentations. Please clarify.

The pMAPK data imply that inhibition of VTA dopamine neurons may have a selective effect on extinction of the tone. It would be good if this could be verified in the freezing data. What was the baseline level of freezing in the context alone in the NpHR group relative to eYFP controls? The period of greatest interest is prior to the first presentation of the tone in the retention test.

Was there some variation in the protocol that produced results in Fig 4? The eYFP rats are different here compared to the previous experiments: there is no evidence of recovery in these rats from the end of extinction to the retention test. In the absence of this recovery, one is led to draw inferences about dopamine projections to the NAc mShell in extinction. If this recovery had occurred, one would be led to a different set of inferences about dopamine projections to the core. Were the three groups of rats run at the same time in this experiment? This would somewhat allay a lingering concern over the interpretation of these data.

This notwithstanding, there seems to be an error in the labelling of the figure, where it is claimed "VTA

dopamine neurons projecting to NAc mShell facilitate extinction memory formation." The data don't support this interpretation. At best, it could be said that VTA dopamine neurons projecting to NAc medial shell are required for consolidation of the memory produced by extinction. In fact, the data from this experiment is quite interesting. One could even build the argument that VTA dopamine neurons projecting to the NAc core are required for the learning that occurs in extinction (these rats extinguished at the same rate as controls) but not its consolidation, whereas dopamine neurons projecting the NAc mShell are not required for the learning that occurs in extinction, but are required for its consolidation. (However, the strength of this interpretation rests on some assurances about performance in the eYFP control group).

Page 7, Line 274: "These results also suggest a differential involvement of NAc core vs. mShell in extinction learning, but how dopamine in mShell controls MAPK phosphorylation in vmPFC and amygdala is not clear." The use of the word 'controls' implies something that, again, has not been shown here. It would be better to use the phrase 'relates to.'

It is curious that the discussion notes the importance of phasic versus tonic dopamine signals in regulating extinction, and then continues to speculate in relation to the significance of the present findings for treatment of anxiety disorders. This speculation is not convincing. If retained, it should be noted that, relative to a standard extinction treatment, the available evidence suggests that counterconditioning enhances, rather than reduces, the renewal of extinguished fear with a context shift.

Reviewer #2 (Remarks to the Author):

The authors have responded to the previous criticisms and have conducted new experiments to extend the work in novel directions. I have no further concerns.

Reviewer #3 (Remarks to the Author):

The authors were very responsive to the critiques from all the reviewers. The discovery that VTA projections to NA core and shell differ anatomically and with respect to extinction is important and boosts the impact of the manuscript considerably. While a specific VTA projection mediating within-session extinction was not found, I am satisfied with the caveat regarding as yet unstudied projections.

Author's note: All author replies to Reviewer comments are italicized and major changes to text are highlighted in red

Reviewers' comments:

Reviewer #1 (Remarks to the Author):

A dopaminergic switch for fear to safety transitions

Luo et al

This paper shows that inhibition of VTA dopamine neurons has contrasting effects on fear acquisition and extinction, identified the timing of these effects to the period when an expected US is not presented, and in the case of extinction, identified some of the circuitry through which these neurons exert their influence. The paper has been improved considerably since its original submission: the additional data (especially that collected in relation to fear acquisition) has helped to build a compelling story regarding the role of VTA dopamine neurons in fear learning and memory. It is suitable for publication in Nature Communications once some minor issues have been addressed in the paper.

In relation to the introduction: The final sentence of the first paragraph (excluding abstract) seems to exclude the possibility that the amygdala and/or vmPFC registers the absence of an expected aversive event. Not even the present findings exclude this (or these) possibilities. As such, some constraint should be applied when constructing arguments in this paragraph.

Response: *We did not mean to imply that the omission of expected aversive events could not be encoded in amygdala or mPFC, only that it is known that molecular signaling in mPFC and amygdala are important and that it is not clear what systems detect when expected events are omitted. We've revised the final sentence of this paragraph to be more explicit about what is not known.*

*"While molecular changes occurring in the ventromedial prefrontal cortex (vmPFC) and amygdala are known to be important for storing and consolidating extinction memories^{1,2}, the brain mechanisms for detecting when **an expected aversive event did not occur and** fear responses are no longer appropriate are less well understood."*

Re the analysis used to determine whether "extinction was completely blocked." In Supp Fig 3A, it is claimed that, relative to the first trial of extinction, the level of freezing to the CS was lower at the end of extinction and in the retention test. One presumes that this data/analysis related to Group HpHR only – this should be clarified. While it is clear that freezing declined across extinction and was lower on the final trial relative to the first, it is not at all clear that the level of freezing on the first trial of the retention test was any lower than the first trial of extinction: the overlapping error bars for this comparison are a slight concern. I understand that this is a within-subject comparison, thereby rendering the error-bars unnecessary. Perhaps further justification could be given for selection of the

Holm-Sidak post-hoc test that was used to support this statement? More generally, I don't understand what this analysis has added to the manuscript (at this point or anywhere else where it is repeated). It could be removed without harm to the overall story.

Response: *we understand the Reviewer's sentiment here that this analysis doesn't significantly add to the manuscript's overall message. However, Reviewer 2 requested an analysis like this that would allow us to test the extent to which extinction was suppressed in the experimental groups. Based on this we feel that these analyses should be included. We used the Holm-Sidak test throughout the paper because it gives greater power than other methods³.*

On page 3 (Lines 119+) it is stated: "In eYFP controls, extinction training increased pMAPK levels in lateral amygdala (LA) and infralimbic (IL) subregion of mPFC compared to unpaired controls." What was the unpaired control? The preceding text (and figure) describes a control group that is exposed to the context in the absence of the tone alone presentations. Please clarify.

Response: *we apologize this was a mistake, there was no unpaired control group. This sentence should have referred to the trained-box control animals. We've changed the text to read:*

"In the eYFP treated groups, extinction training increased pMAPK levels in lateral amygdala (LA) and infralimbic (IL) subregion of mPFC compared to chamber exposed controls."

The pMAPK data imply that inhibition of VTA dopamine neurons may have a selective effect on extinction of the tone. It would be good if this could be verified in the freezing data. What was the baseline level of freezing in the context alone in the NpHR group relative to eYFP controls? The period of greatest interest is prior to the first presentation of the tone in the retention test.

Response: *There was minimal pre-CS (before 1st CS) freezing in both groups and no statistically significant difference. We have included this analysis in the revised text:*

"There were no detectable differences between eYFP and NpHR animals in freezing responses prior to the onset of the first CS (% freezing in 20 sec before 1st CS onset: NpHR=2.7+/-1.8% SEM, eYFP=5.6+/-2.0% SEM; p=0.33, student's T-test)."

Was there some variation in the protocol that produced results in Fig 4? The eYFP rats are different here compared to the previous experiments: there is no evidence of recovery in these rats from the end of extinction to the retention test. In the absence of this recovery, one is led to draw inferences about dopamine projections to the NAc mShell in extinction. If this recovery had occurred, one would be led to a different set of inferences about dopamine projections to the core. Were the three groups of rats run at the same time in this experiment? This would somewhat allay a lingering concern over the interpretation of these data.

This notwithstanding, there seems to be an error in the labelling of the figure, where it is claimed "VTA dopamine neurons projecting to NAc mShell facilitate extinction memory formation." The data don't support this interpretation. At best, it could be said that VTA dopamine neurons projecting to NAc

medial shell are required for consolidation of the memory produced by extinction. In fact, the data from this experiment is quite interesting. One could even build the argument that VTA dopamine neurons projecting to the NAc core are required for the learning that occurs in extinction (these rats extinguished at the same rate as controls) but not its consolidation, whereas dopamine neurons projecting the NAc mShell are not required for the learning that occurs in extinction, but are required for its consolidation. (However, the strength of this interpretation rests on some assurances about performance in the eYFP control group).

Response: *The Reviewer makes an astute observation here about the data, there is some consolidation in the eYFP controls (and possibly core ArchT group) this is absent in the mShell group. The procedures were identical to the other experiments and all groups were run together, so this doesn't explain the subtle differences in the behavioral results across different experiments in the paper. Variability in behavior is apparent across all behavioral experiments and we believe this is inherent in behavioral experiments in general. Although as experimenters we seek to control all conditions, there are always some which are more difficult to control (weather, time of year, etc). We do agree with the Reviewer that using 'memory formation' is not completely accurate and have removed this phrasing or, in the offending figure legend, changed it to:*

"VTA dopamine neurons projecting to NAc mShell facilitate consolidation of extinction learning."

Page 7, Line 274: "These results also suggest a differential involvement of NAc core vs. mShell in extinction learning, but how dopamine in mShell controls MAPK phosphorylation in vmPFC and amygdala is not clear." The use of the word 'controls' implies something that, again, has not been shown here. It would be better to use the phrase 'relates to.'

Response: *this has been changed in the revised manuscript:*

"These results also suggest a differential involvement of NAc core vs. mShell in extinction learning, but how dopamine in mShell relates to MAPK phosphorylation in vmPFC and amygdala is not clear."

It is curious that the discussion notes the importance of phasic versus tonic dopamine signals in regulating extinction, and then continues to speculate in relation to the significance of the present findings for treatment of anxiety disorders. This speculation is not convincing. If retained, it should be noted that, relative to a standard extinction treatment, the available evidence suggests that counterconditioning enhances, rather than reduces, the renewal of extinguished fear with a context shift.

Response: *We have added a reference to the manuscript regarding the enhanced renewal⁴. We would also point out though that there are counterconditioning procedures which do not produce this increase in renewal and, in fact, reduce renewal⁵.*

Reviewer #2 (Remarks to the Author):

The authors have responded to the previous criticisms and have conducted new experiments to extend the work in novel directions. I have no further concerns.

Reviewer #3 (Remarks to the Author):

The authors were very responsive to the critiques from all the reviewers. The discovery that VTA projections to NA core and shell differ anatomically and with respect to extinction is important and boosts the impact of the manuscript considerably. While a specific VTA projection mediating within-session extinction was not found, I am satisfied with the caveat regarding as yet unstudied projections.

1. Herry, C. et al. Neuronal circuits of fear extinction. *Eur J Neurosci* **31**, 599-612 (2010).
2. Quirk, G.J. & Mueller, D. Neural mechanisms of extinction learning and retrieval. *Neuropsychopharmacology* **33**, 56-72 (2008).
3. Seaman, M.A., Levin, J.R. & Serlin, R.C. New Developments in pairwise multiple comparisons: Some powerful and practicable procedures. *Psychological Bulletin* **110**, 577-586 (1991).
4. Holmes, N.M., Leung, H.T. & Westbrook, R.F. Counterconditioned fear responses exhibit greater renewal than extinguished fear responses. *Learn Mem* **23**, 141-50 (2016).
5. Thomas, B.L., Cutler, M. & Novak, C. A modified counterconditioning procedure prevents the renewal of conditioned fear in rats. *Learning & Motivation* **43**, 24-34 (2012).

REVIEWERS' COMMENTS:

Reviewer #1 (Remarks to the Author):

The authors have addressed all of the points raised in the second review. The manuscript will make an excellent contribution to the literature. It is ready for publication in its present form.